# Workflow Discovery from Dialogues in the Low Data Regime

**Amine El Hattami**                                    *amine.elhattami@servicenow.com*
*ServiceNow Research*
*Polytechnique Montréal, Montréal, Canada*

**Issam Laradji**
*ServiceNow Research*

**Stefania Raimondo**
*ServiceNow Research*

**David Vázquez**
*ServiceNow Research*

**Pau Rodriguez**
*ServiceNow Research*

**Christopher Pal**
*ServiceNow Research*
*Polytechnique Montréal, Montréal, Canada*

**Reviewed on OpenReview:** `https://openreview.net/forum?id=L9othQvPks`

## Abstract

Text-based dialogues are now widely used to solve real-world problems. In cases where solution strategies are already known, they can sometimes be codified into *workflows* and used to guide humans or artificial agents through the task of helping clients. We introduce a new problem formulation that we call Workflow Discovery (WD) in which we are interested in the situation where a formal workflow may not yet exist. Still, we wish to discover the set of actions that have been taken to resolve a particular problem. We also examine a sequence-to-sequence (Seq2Seq) approach for this novel task. We present experiments where we extract workflows from dialogues in the Action-Based Conversations Dataset (ABCD). Since the ABCD dialogues follow known workflows to guide agents, we can evaluate our ability to extract such workflows using ground truth sequences of actions. We propose and evaluate an approach that conditions models on the set of possible actions, and we show that using this strategy, we can improve WD performance. Our conditioning approach also improves zero-shot and few-shot WD performance when transferring learned models to unseen domains within and across datasets. Further, on ABCD a modified variant of our Seq2Seq method achieves state-of-the-art performance on related but different problems of Action State Tracking (AST) and Cascading Dialogue Success (CDS) across many evaluation metrics. [1]

---

[1]Code available at https://github.com/ServiceNow/workflow-discovery

# 1    Introduction

Task-oriented dialogues are ubiquitous in everyday life and customer service in particular. Customer support agents use dialogue to help customers shop online, make travel plans, and receive assistance for complex problems. Behind these dialogues, there could be either implicit or explicit *workflows* – actions that the agent has followed to ensure the customer request is adequately addressed.

For example, booking an airline ticket might comply with the following workflow: pull up an account, register a seat, and request payment. Services with no formal workflows struggle to handle variations in how a particular issue is resolved, especially for cases where customer support agents tend to follow "unwritten rules" that differ from one agent to another, significantly affecting customer satisfaction and making training new agents more difficult. However, correctly identifying each action constituting a workflow can require significant domain expertise, especially when the set of possible actions and procedures may change over time. For instance, newly added items or an update to the returns policy in an online shopping service may require modifying the established workflows.

In this work, we focus on "workflow discovery" (WD) – the extraction of workflows that have either implicitly or explicitly guided task-oriented dialogues between two people. Workflows extracted from a conversation consist of a summary of the key actions taken during the dialogue. These workflows consist of pre-defined terms for actions and slots when possible, but our approach also allows for actions that are not known to be invented by the model online and used as new steps in the generated workflow. Our approach is targeted toward the task of analyzing chat transcripts between real people, extracting workflows from transcripts, and using the extracted workflows to help design automated dialogue systems or to guide systems that are already operational. Alternatively, extracted workflows might also be used for human agent training. One might imagine many scenarios where an analyst might use WD to understand if an unresolved problem is due to a divergence

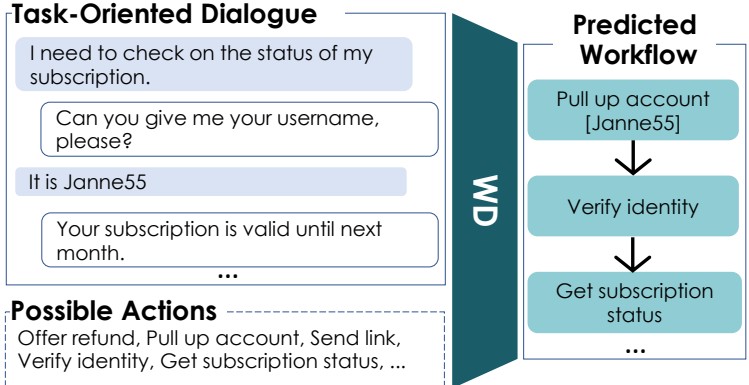

Figure 1: Our method for extracting workflows from dialogues. The input consists of the dialogue utterances and the list of possible actions if available. The output is the workflow' followed to resolve a specific task, consisting of the actions (e.g., Verify identity) and their slot values (e.g., Janne55).

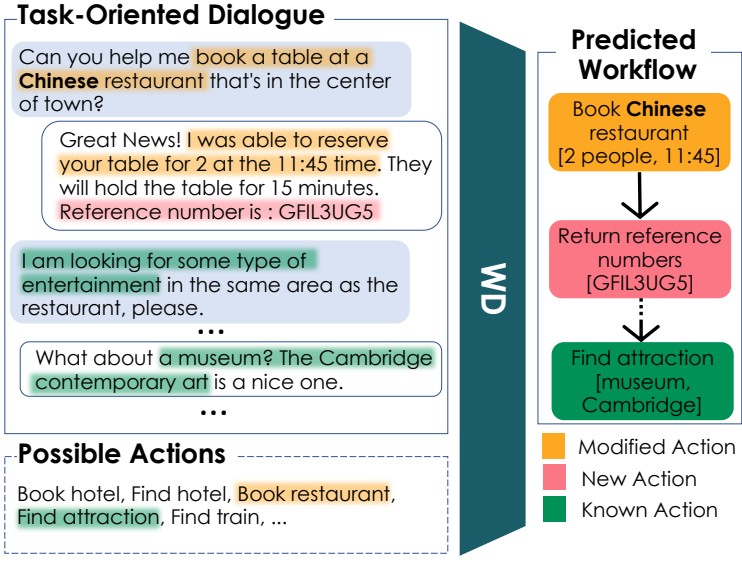

Figure 2: Our conditioning approach allows for better zero-shot and few-shot performance. Entirely new workflow actions (i.e, not in the list of possible actions) can be proposed, as well as those based on minor modifications to known actions.

from a formal workflow or to understand if workflows have organically emerged, even in cases where no formal workflow documentation exists. WD is related to the well-established field of process mining, but process mining typically extracts workflow information from event logs as opposed to unstructured dialogues encoded as natural language. Our work shows that traditional process mining can be dramatically expanded using modern NLP and sequence modeling techniques, as we propose and explore here. Our approach correspondingly allows traditional event logging information to be combined with NLP allowing systems constructed using our approach to benefit from recent advances in large language models (LLMs), including enhanced zero-shot and few-shot learning performance.

Our WD problem formulation here is related to dialogue act modeling in the sense that different steps along our workflows may be composed of dialogue acts in the sense of Stolcke et al. (2000), such as: posing yes or no questions, acknowledging responses, thanking, etc. Since WD is performed offline and focuses on workflow extraction, more fine-grained dialogue acts can help guide a WD model to extract more abstract workflow steps. WD is also different from standard (closed world) intent detection and slot filling Liu & Lane (2016), dialogue state tracking (DST) Williams et al. (2014) and open world intent and slot induction Perkins & Yang (2019) in part because WD focuses on extracting actions taken by an agent as opposed to intents of the user. We explore WD model variants that extract both agent actions and arguments or slots obtained from the user. WD could be seen as a generalization of the recently proposed notion of Action State Tracking (AST) (Chen et al., 2021), but WD differs from AST in that: 1) WD is performed offline summarizing dialogues in terms of workflows; 2) unlike AST, WD does not require actions to be annotated at the turn level; 3) unlike AST, WD doesn't require known pre-defined actions and slots. One of the uses of WD is to create new names for new actions on the fly along with new slot types and extracted values; and, 4) these features of WD allow it to extract workflows containing new action and slot types which differ significantly from action and slot sequences seen during training. We summarize these differences between AST and WD in Table 1, and we discuss all the relationships of WD to prior work in more detail in our Related Work section below.

Table 1: A summary of the differences between AST and WD.

|  | AST | WD |
| --- | --- | --- |
| Performed offline | No | Yes |
| Requires annotated action turns | Yes | No |
| Requires known actions and slots types | Yes | No |
| Significant deviation from known actions and slots possible | No | Yes |

To address the challenges of WD introduced by dynamically changing actions, distributional shifts, and the challenges of transferring to completely new domains with limited labeled data, we propose a text-to-text approach that can output the workflow consisting of the full set of actions and slot values from a task-oriented dialogue. Further, to better adapt to unseen domains, we condition our method on the full or partial set of possible actions as shown in Figure 2, enabling us to perform zero-shot and few-shot learning. Figure 2 also illustrates the type of generalization possible with our approach in these regimes where we can propose new actions never seen during training. We investigate four scenarios for extracting workflows from dialogues under differing degrees of distributional shift: (1) *In-Domain workflows*, where all workflow actions have been seen during training. (2) *Cross-domain zero-shot*, where actions from selected domains have been omitted during training but are present in the valid and test sets, (3) *Cross-dataset zero-shot*, where a model trained on all the domains of one dataset is applied to another domain in a zero-shot setting, and (4) *Cross-dataset few-shot*, where a model is trained on all the domains of one dataset, then trained on another domain in a few-shot setting.

Our contributions can be summarized as follows:

- We propose a new formulation to the problem of workflow extraction from dialogues, which, to our knowledge, has not been examined or framed as we present here. We cast the problem as summarizing dialogues with workflows and call the task Workflow Discovery (WD).

- We propose a text-to-text approach to solve the WD task that takes entire dialogues as input and outputs workflows consisting of sequences of actions and slot values that can be used to guide future dialogues. We test our approach using various state-of-the-art text-to-text models and show its efficacy on the Action Based Conversations Dataset (ABCD) (Chen et al., 2021).

- We propose a conditioning mechanism for our text-to-text approach, providing the model with a set of possible actions to use as potential candidates. We show that this mechanism allows our method to be used in dramatically different domains from the MultiWOZ dataset (Budzianowski et al., 2018), yielding good cross-dataset zero-shot workflow extraction performance. Moreover, it allows an important performance increase in the cross-dataset few-shot setting.

- Using a variant of our approach, we achieve state-of-the-art results on related but different and more standard tasks of Action State Tracking (AST) and Cascading Dialogue Success (CDS) on the ABCD evaluation.

## 2 Related Work

Our work intersects with three major groups of topics: Task-Oriented Dialogues, Sequence-to-Sequence Text Generation, Intent/Slot induction, Action State Tracking (AST) and Process Mining & Discovery.

**Task-Oriented Dialogues** In task-oriented dialogues, the system must grasp the users' intentions behind their utterances since this is the basis for selecting an appropriate system action, it should be able to understand and respond to a wide range of user inputs, as well as handle complex tasks and maintain coherence in the dialogue Stolcke et al. (2000); Henderson et al. (2014). The two key tasks are intent detection and named entity identification, both of which have comparable components in task-oriented dialogues. Intent detection may be viewed as a classification task (action classification), with user utterances allocated to one or more intent categories. The goal of named entity recognition is to categorize entities in a given utterance, such as hotel names, room characteristics, and attribute values Zang et al. (2020).

Although most systems perform intent detection and named entity identification for generic conversation systems Rafailidis & Manolopoulos (2018); Yan et al. (2017), the main goal of our method is to output a set of actions that resolve the task specified in the dialogue that abide by the system's guidelines.

**Intent and Slot Induction.** In human-human dialogues, the purpose of intent/slot induction is to determine user intents from user query utterances. In more recent years, there have been several efforts on intent/slot detection in conversation systems based on methods like capsule networks and relation networks Min et al. (2020); Qin et al. (2020); Zhang et al. (2018); Qin et al. (2019); Niu et al. (2019). However, they all make the assumption that the intents are within a closed world in that the intents in the test set also appear in the training set. In contrast, while our work is about identifying the set of actions that address the task in a dialogue we consider the possibility that the actions in the test set are novel.

Prior work has also sought to identify novel intents and slots. For example, Brychcín & Král (2016) proposed an unsupervised method for identifying intents without utilizing prior knowledge of known intents. Ryu et al. (2018) and Yu et al. (2017) proposed methods based on adversarial learning, whereas Kim & Kim (2018) proposed methods based on an in-domain classifier and an out-of-domain detector to detect unknown intents. Perkins & Yang (2019) used clustering with multi-view representation learning for intent induction, whereas Zeng et al. (2021) proposed a method based on role-labeling, concept-mining, and pattern-mining for discovering intents on open-domain dialogues. Closer to our work is Yu et al. (2022) which leveraged large language models for discovering slot schemas for task-oriented dialog in an unsupervised manner. They use unsupervised parsing to extract candidate slots followed by coarse-to-fine clustering to induce slot types. In contrast, in our work, we use large language models and a prompting technique to detect and even label out-of-distribution actions. Rather than clustering embeddings or other structures, we leverage knowledge obtained during transformer pre-training, and a special prompting scheme to let the LLM simply generate new workflow steps as text. Our approach allows our models to propose new, plausible, and typically quite

appropriate names for workflow steps that have not been seen in the workflow extraction training data. We present our analysis of these types of experiments in Section 6.3.3.

**Sequence-to-Sequence models**   Text generation has emerged as one of the most significant but difficult problems in natural language processing (NLP). RNNs, CNNs, GNNs, and pretrained language models (PLMs) have enabled end-to-end learning of semantic mappings from input to output  (Raffel et al., 2020b; Lewis et al., 2019). The architectures of PLMs are divided into Encoder-Decoder (like BERT Devlin et al. (2018) and T5) and Decoder-only (like GPT Brown et al. (2020)) architectures. We employ Encoder-Decoder-based sequence-to-sequence models in this study since we want to accept an entire dialogue as input and produce a set of actions as workflow. Many Encoder-Decoder methods exist in the literature and some of their main differences are how they were pretrained. For instance, PEGASUS Zhang et al. (2020) trains by masking/removing important sentences from an input document, T5 Raffel et al. (2020b) uses a multi-task approach for pretraining, BART Lewis et al. (2019) uses static masking of sentences across epochs, and RoBERTa Liu et al. (2019) uses dynamic masking, different parts of the sentences are masked across epochs. These types of architectures form the basis of our evaluation.

Our work is connected to models developed for translation (Sutskever et al., 2014) and summarization (Gliwa et al., 2019; Zhang et al., 2020), structured text generation such as Text-to-Code Wang et al. (2021) and Text-to-SQL Scholak et al. (2021). However, here we focus on a new task, where dialogues are transformed into sequences of steps that an agent has used to solve a task.

**Structured Text Generation.**   Structured text generation such as text-to-SQL and text-to-code have received a lot of recent attention Warren & Pereira (1982); Zettlemoyer & Collins (2012); Finegan-Dollak et al. (2018); Feng et al. (2020); Scholak et al. (2021); Wang et al. (2021); Yu et al. (2019). The majority of the past text-to-SQL effort has been devoted to transforming a single, complicated question into its accompanying SQL query. Only a few datasets have been created to link context-dependent inquiries to structured queries, but the recent conversational text-to-SQL (CoSQL) work of Yu et al. (2019) has examined the generation of SQL queries from multiple turns of a dialogue. Their work also involved creating a dataset under a WOZ setting involving interactions between two parties. This dataset consists of diverse semantics and discourse that encompass most sorts of conversational DB querying interactions; for example, the system will ask for clarification of unclear inquiries or advise the user of irrelevant questions. While this dataset is a great benchmark for SQL generation, here we are specifically interested in transforming dialogues into workflows consisting of much more diverse types of interactions. Correspondingly, we focus on the recently introduced ABCD dataset Chen et al. (2021), which consists of dialogues where an agent's actions must accommodate both the desires expressed by a customer and the constraints set by company policies and computer systems. Our goal is to predict sequences of agent actions and the arguments for those actions, which are often obtained from user-provided information.

**AST and CDS for Task-Oriented Dialogues.**   Action State Tracking (AST) is a task proposed by Chen et al. (2021) which tries to predict relevant intents from customer utterances while taking Agent Guidelines into consideration, which goes steps beyond traditional dialog state tracking (DST) Lee et al. (2021). For instance, a customer's utterance might suggest that the action is to update an order. However, following the agent's guidelines, the agent might need to perform other actions prior to addressing the customer's intent (e.g., validating the customer's identity). While AST shares some similarities with traditional DST tasks, its main advantage is that it parses both customer intents and agent guidelines to predict agent actions. However, AST relies on annotated action turns making it difficult to use on existing dialogue datasets without substantial annotation effort. Further, AST relies on agent guidelines which require prior knowledge of all possible actions.  Chen et al. (2021) also proposed the Cascading Dialogue Success (CDS) task that aims to access the model's ability to predict actions in context. CDS differs from AST since it does not assume that an action should occur in the current turn, but adds the possibility to take an action, respond with an utterance (e.g., requesting information from the customer), or terminate the conversation when the agent completed the task. Further, CDS is measured using a modified version of the cascading evaluation

metric that accesses success over successive turns as opposed to AST which measures success in isolation on a single turn.

## 3 Workflow Discovery (WD)

### 3.1 WD Task Definition

We propose a novel task we call Workflow Discovery (WD) to extract the actual workflow followed by an agent from a task-oriented dialogue. A workflow is a *sequence* of *actions* with their respective slot values in the same order in which they appeared during the conversation. Formally, we define the WD task as follows:

*Given a dialogue $D = \{u_1, u_2, ..., u_n\}$, where $n$ is the total number of utterances in the dialogue, and an optional list of possible actions $\delta = \{a_1, ..., a_z\}$, where $z$ is the number of known actions, and $a_z$ is a unique workflow action. A model should predict the target workflow $W = \{(a_1, \{v_1^j | 0 <= j <= n_1\}), ..., (a_k, \{v_k^i | 0 <= j <= n_k\})\}$, where $a_i \in \delta$, $v_i^j$ is the $j^{th}$ slot value and $n_i$ is the number of available slot values for action $a_i$, and $k$ is the number of workflow actions. Further, the model should also be able to formulate new compound keywords to characterize new actions as well as extract their slot values for actions that are not a part of the known action domain.*

### 3.2 WD Compared to Existing Tasks

Workflow Discovery differs from dialogue state tracking (DST) where in WD, we are interested in extracting the sequence of actions that have been followed to resolve an issue. We are particularly interested in the situation where the actions that are needed to resolve a problem associated with a given intent or task *are not known a priori* and must be invented by the WD model. This is in sharp contrast to DST which generally requires dialogue states to be known and pre-defined. DST also generally focuses on tracking the state of one party in the conversation, e.g., the user. In contrast, WD extracts actions executed by the agent and slot values collected from user utterances. Furthermore, WD differs from Action State Tracking (AST), since WD aims to extract *all* actions and their matching slot values, using *all* dialogue utterances without any extra metadata like the annotated action turns at once (no online tracking). In contrast, AST predicts the next action at a given turn, given the previous turns only. Models trained for AST help agents select the next action following the agent guidelines. In contrast, models trained for WD extract all actions from chat logs between real people, including those that might deviate from the agent guidelines. WD is aimed at agent training or workflow mining with the goal of formalizing the discovered processes for subsequent automation. WD can be applied to completely new, organic tasks where possible actions are unknown. Some aspects of AST could be seen as a subset of WD when actions and slots are known. However, we believe that WD is more closely related to the concept of summarization using a specialized vocabulary for actions and slots, when possible, but inventing new terms when needed.

We compare and contrast WD, AST, and DST in Figure 9. We note that Figure 9a shows a situation where our WD approach has succeeded in inventing a new term (i.e., "Generate new code") which describes the exact action the agent performed. This result matches the agent guidelines (Chen et al., 2021), where there is a clear distinction between offering a promo code when a customer is unhappy in which our approach uses the known "promo code" action, and generating a new code since the old one is no longer working. A good WD method should exhibit compositional generalization when creating new terms for actions and their arguments. For example, if a system has been trained to summarize dialogues that involve granting a refund for ordering a pair of pants, the WD summary of a new dialogue involving the refund of a shirt should use a vocabulary consistent with the manner in which the refund of a pair of pants has been expressed.

Finally, in contrast to policy learning, WD aims to extract a workflow from dialogue logs describing the sequence of agent actions that depend on slot values extracted from user utterances. The extracted sequence may differ significantly from the actions of an optimal or estimated policy. WD focuses on extracting potentially out-of-distribution workflows, where new action names and slot values derived from new domains must be invented. This makes WD very different from policy learning and more similar to dialogue policy

act extraction, but where new acts and slot values must be invented on the fly to summarize new concepts at test time.

## 4 Baselines and Methodology

In this section, we describe our methodology for the WD task. Moreover, since there is no existing baseline for our novel WD task, we included our text-to-text variants for both the AST and CDS tasks that we used to test our text-to-text task casting scheme against existing benchmarks.

### 4.1 Text-to-Text Workflow Discovery

We cast the WD task as a text-to-text task where the input of the model $P_{WD}$ consists of all utterances and the list of possible actions, if available or partially available, formatted as

$$P_{WD} = \text{Dialogue: } u_1, ..., u_n \text{ Actions: } a_1, ..., a_z$$

where $u_n$ is a dialogue utterance, $n$ is the total number of utterances in the dialogue, $a_z$ is a workflow action, and $z$ is the total number of available actions. "$Dialogue$:" and "$Actions$:" are delimiters that separate the dialogue utterances from the list of actions. Further, we omit the "$Actions$:" delimiter when possible actions are not provided. Adding the possible actions to the input can be seen as a way to condition the model to use the provided actions rather than invent new ones. During training, the possible actions added after "$Actions$:" in the input is a *shuffled* set comprised of the current sample target actions and a randomly chosen number $r_{min} <= r <= z$ of actions, where $r_{min}$ is a hyper-parameter. This technique aims to make the model invariant to the position and the number of workflow actions, especially when adapting to a new domain in the zero-shot and few-shot settings. For all other settings (i.e., during validation, testing, or the zero-shot setting), the list of possible actions contains *all* known actions without any modification. Finally, We use the source prefix "$Extract\ workflow$:" The target workflow $T_{WD}$ is formatted as

$$T_{WD} = a_1[v_1^1, ..., v_1^{n_1}]; ...; a_k[v_k^1, ..., v_k^{n_k}]$$

where $a_k$ is a workflow action, $k$ is the number of actions, $v_k^{n_k}$ is a slot value, and $n_k$ is the number of slot values for action $k$. "[" and "]" encapsulate the slot values. "," and ";" are the separators for the slot values and actions. Moreover, if an action has no slot values, the target slot value list is set explicitly to [*none*]. An example of this casting scheme is shown in Figure 9a.

Our text-to-text framework differs from other work since our prompts don't include slot descriptions or value examples compared to Zhao et al. (2022a). Moreover, Adding lists of possible actions to the prompt makes our method more flexible in the zero-shot setup (allowing new actions to be added on-the-fly) and improves performance in the few-shot setup. Finally, we don't prefix utterances with speaker prefixes (e.g., "User: ") compared to Zhao et al. (2022a); Lin et al. (2021a), making our technique usable when this information is unavailable or inaccurate; for example, in cases where the chat transcript originated from an upstream text-to-speech model.

### 4.2 Text-to-Text Action State Tracking

The goal of the AST task is to predict a *single* action and its slot values given only the previous turns. This task is similar to the traditional DST with the difference that agent guidelines constrain the target actions. For example, an utterance might suggest that the target action is "validate-purchase", but the agent's proposed gold truth is "verify-identity" per the agent guideline because an agent needs to verify a customer's identity before validating any purchase.

We cast the AST task as a text-to-text task where the input of the model $P_{AST}$ consists of all the utterances formatted as

$$P_{AST} = \text{Context: } u_1, ..., u_t$$

where $u_t$ is a dialogue turn, including action turns, and $t$ is the index of the current turn. "Context:" is a delimiter. We use the source prefix "$Predict\ AST$:". The target $T_{AST}$ is formatted as

$$T_{AST} = a_t[v_t^1, ..., v_t^m]$$

where $a_t$ is an action, $v^m$ is a slot value, and $m$ is the number of slot values. "[" and "]" encapsulate the slot values. "," is the separator for the slot values. Moreover, if an action has no slot values, the target slot value list is set explicitly to $[none]$. An example of this casting scheme is shown in Figure 9b.

### 4.3 Text-to-Text Cascading Dialogue Success

The CDS task aims to evaluate a model's ability to predict the dialogue intent and the next step given the previous utterances. The next step can be to take action, respond with a text utterance, or end the conversation. Moreover, the CDS task is evaluated over successive turns. In contrast, AST assumes that the current turn is an action turn and is evaluated over individual turns.

We cast the CDS task as a text-to-text task where the model input $P$ is formatted as

$$P_{CDS} = \text{Context:} u_1, ..., u_t \text{ Candidates:} c_1, ..., c_v$$

where $u_t$ is a dialogue turn, including action turns, and $t$ is the index of the current turn. $c_v \in C_t$ is a text utterance from the current agent response candidate list $C_t$, and $v$ is the size of $C_t$. "Context:" and "Candidates:" are delimiters. Finally, We use the source prefix "$Predict\ CDS$:". The target $T_{CDS}$ is formatted differently depending on the target next step. When the next step is to take an action, the target is formatted as follows

$$T_{CDS}^{action} = i;\ action;\ a_t[v_t^1, ..., v_t^m]$$

where $i$ is the dialogue intent, and values $a_t[v_t^1, ..., v_t^m]$ is the step name and slots similar to the AST task formulation above. When the next step is to respond, the target is formatted as follows

$$T_{CDS}^{respond} = i;\ respond;\ c_{t+1}$$

where $c_{t+1} \in C_t$ is the expected response utterance. When the next step is to terminate the dialogue, the target is formatted as follows

$$T_{CDS}^{end} = i;\ end$$

An example of this casting scheme is shown in Figure 9c.

## 5 Experimental Setup

### 5.1 Implementation Details

In our experimentation, we used the T5 (Raffel et al., 2020b), BART (Lewis et al., 2019), and PEGASUS (Zhang et al., 2020) models for the WD task, which we call WD-T5, WD-BART, and WD-PEGASUS, respectively. Furthermore, we use a T5 model for the text-to-text AST and CDS tasks, which we call, AST-T5, and CDS-T5, respectively. We use the Huggingface Transformers Pytorch implementation[2] for all model variants and use the associated summarization checkpoints[3] fine-tuned on CNN/DailyMail (Hermann et al., 2015) for all models. For T5, we use the small (60M parameters), base (220M parameters), and large (770M parameters) variants, and the large variant for both BART (400M parameters) and PEGASUS (569M parameters). We fine-tuned all models on the WD tasks for 100 epochs for all experiments with a learning rate of 5e-5 with linear decay and a batch size of 16. We set the maximum source length to 1024 and the maximum target length to 256. For the BART model, we set the label smoothing factor to 0.1. We fine-tuned AST-T5, and CDS-T5 for 14 and 21 epochs, respectively, matching the original work of Chen et al. (2021), and used similar hyper-parameters used for the WD task. In all experiments, we use $r_{min} = 10$ as described in Section 4.1. Finally, We ran all experiments on 4 NVIDIA A100 GPUs with 80G memory, and the training time of the longest experiment was under six hours.

---

[2]https://huggingface.co/transformers
[3]https://huggingface.co/models

## 5.2 Datasets

**ABCD** (Chen et al., 2021) contains over 10k human-to-human dialogues split over 8k training samples and 1k for each of the eval and test sets. The dialogues are divided across 30 domains, 231 associated slots, and 55 unique user intents from online shopping scenarios. Each intent requires a distinct flow of actions constrained by agent guidelines. To adapt ABCD to WD, we choose agent actions to represent the workflow actions. From each dialogue, we create the target workflow by extracting the actions and slot values from each action turn in the same order they appear in the dataset. Then, we remove all action turns from the dialogue keeping only agent and customer utterances. We use the same data splits as in ABCD. Furthermore, we use natural language action names (i.e., using "pull up customer account" instead of "pull-up-account"), and we report the results of an ablation study in Section 6.3.5 showing the benefits of using natural language action names. Table 2 shows a subset of the names we used, and the full list can be found in Appendix A.4.2.

Table 2: Subset of ABCD workflow actions

| Action Name | Natural Language Action Name |
| --- | --- |
| pull-up-account | pull up customer account |
| enter-details | enter details |
| verify-identity | verify customer identity |

**MultiWOZ** (Budzianowski et al., 2018) contains over 10k dialogues with dataset splits similar to ABCD across eight domains: *Restaurent, Attraction, Hotel, Taxi, Train, Bus, Hospital, Police*. We use MultiWOZ 2.2 (Zang et al., 2020) as it contains annotated per turn user intents and applies additional annotations correction similar to Ye et al. (2022). In the MultiWOZ dataset, customer intents represent the workflow actions. We assume that the system will always perform a customer intent. To MultiWoz to the WD task, we create the target workflow by extracting the set of intents and slot values in the same order they appear in the dialogue. We don't make any modifications to the dialogue utterance since MutliWoz does not include action turns or any extra metadata. Similar to ABCD, we use natural language action names. Table 3 shows a subset of the natural language action names. See Appendix A.4.1 for the full list.

Table 3: Subset of MultiWOZ workflow actions

| Action Name | Natural Language Action Name |
| --- | --- |
| find_hotel | search for hotel |
| find_train | search for train |
| book_restaurant | book table at restaurant |

Our public code repository[4] contains all the source code necessary to create the WD dataset from ABCD and MultiWoz. The repository also contains the annotated ABCD test subset for human evaluation. Further, we report dataset statistics in Appendix A.4.3.

## 5.3 Metrics

We evaluate the WD task using the Exact Match (EM) and a variation of the Cascading Evaluation (CE) (Suhr et al., 2019) metrics similar to Chen et al. (2021). The EM metric only gives credit if the predicted workflow (i.e., actions and slot values) matches the ground truth. However, it is not a good proxy of model performance on the WD task due to the sequential nature of workflows. For example, a model that predicts all workflow actions correctly but one will have an EM score of 0, similar to another model that predicted all actions wrong. In contrast, the CE metric gives partial credit for each successive correctly predicted action and its slot values. Nonetheless, we kept the EM metric for applications where predicting the exact workflow is critical. Furthermore, we report an action-only Exact Match (Action-only EM) and Cascading

---

[4]https://github.com/ServiceNow/workflow-discovery

Evaluation (Action-only CE) for some experiments to help isolate the task complexity added by predicting the slot values. Finally, due to the text-to-text nature of our approach, we use stemming, and we ignore any failure that occurs due to a stop word miss-match (e.g., we assume that *the lensfield hotel* is equivalent to *lensfield hotel*). Moreover, we use BERTScore (Zhang* et al., 2020) to evaluate the action in all zero-shot experiments. We assume that a predicted action is valid if it has an F1-score above 95% (e.g., *check customer identity* is equivalent to *verify customer identity*).

We evaluate the AST task similar to Chen et al. (2021) where B-Slot and Value are accuracy measures for predicting action names and values, respectively. Action is a joint accuracy for both B-Slot and Value metrics. Furthermore, We evaluate the CDS task similar to Chen et al. (2021) using the Cascading Evaluation (CE) metric that relies on the Intent, NextStep, B-Slot, Value, and Recall@1 metrics that are accuracy measures for the dialogue intent, next step, action, value, and next utterance predictions. We omit the Recall@5, and Recall@10 metrics results since our model only predict a single next utterance instead of ranking the top 5 or 10 utterances. However, the cascading evaluation calculation result remains valid since it only uses the Recall@1 scores.

## 6 Experimental Results

To validate the efficacy of our approach against existing baselines, we first compare our results to the current state-of-the-art on the AST and CDS tasks and show that our method achieves state-of-the-art results on both tasks. Then, we describe the experiments we used to evaluate our approach on the WD task, report the results, and discuss their conclusions.

### 6.1 Action State Tracking

Following the same evaluation method used in Chen et al. (2021), we compared our AST-T5 model against ABCD-RoBERTa (Chen et al., 2021), the current state-of-the-art on the AST task. We report the results in Table 4. Moreover, we only report the results of AST-T5-Small (60M parameters) variant for a fair comparison since the ABCD-RoBERTa model is around 125M parameters while our AST-T5-Base is 220M parameters.

Table 4: Our AST-T5-Small results on the AST task using the ABCD test set.

| Params | Models | B-Slot | Value | Action |
|--------|--------|--------|-------|--------|
| 124M | ABCD-RoBERTa | **93.6%** | 67.2% | 65.8% |
| 60M | AST-T5-Small (Ours) | 89.1% | **89.2%** | **87.9%** |

Our AST-T5-Small variant achieves state-of-the-art results on the AST task while using 50% less trainable parameters. Furthermore, our text-to-text approach is easily adaptable to any new domain without any model change. In contrast, an encoder-only approach like the one used by ABCD-RoBERTa requires an update to the classification head to add a new action or slot value.

### 6.2 Cascading Dialogue Success (CDS)

Following the same evaluation method used in Chen et al. (2021), we compared our CDS-T5 models against the current state-of-the-art on the CDS task. In Table 5, we report the best results (ABCD Ensemble) for each metric from Chen et al. (2021).

Our CDS-T5-Small (60M parameters) outperforms the current state-of-the-art (with up to 345M parameters) while using 80% less trainable parameters. Furthermore, Our CDS-T5-Base achieves a new state-of-the-art on the CDS task while using 36% fewer trainable parameters, showing the advantage of our text-to-text approach. Specifically, our CDS-T5-Base outperformed the ABCD Ensemble on the next utterance prediction (recall@1) by 27.4 points. Furthermore, both our models scored exceptionally on predicting the next step.

Table 5: CDS-T5 results on the CDS task using the ABCD test set. Intent, NextStep, B-Slot, Value, Recall@1 are accuracy measures for the dialogue intent, next step, action, value, and next utterance predictions, respectively. CE is the cascading evaluation that uses all other metrics. ABCD Ensemble is the ensemble with the best scores from ABCD.

| Params | Models | Intent | Nextstep | B-Slot | Value | Recall@1 | CE |
|--------|--------|--------|----------|--------|-------|----------|-----|
| 345M | ABCD Ensemble | **90.5%** | 87.8% | 88.5% | 73.3% | 22.1 | 32.9% |
| 60M | CDS-T5-Small (Ours) | **85.7%** | 99.5% | 85.9% | 75.1% | 40.7 | 38.3% |
| 220M | CDS-T5-Base (Ours) | 86.0% | **99.6%** | 87.2% | **77.3%** | **49.5** | **41.0%** |

Finally, CDS-T5-Base outperforms the human performance (Chen et al., 2021) on value accuracy by 1.8 points.

### 6.3 Workflow Discovery (WD)

This section shows the experiments we performed to explore the WD task using several baseline models. First, we report the results in the following learning setups: in-domain, cross-domain zero-shot, cross-dataset zero-shot, and few-shot. Then, report the results of the experiments showing the performance improvement when using natural language action names and using models fine-tuned on summarization. It is worth noting that the idea behind using multiple model architectures and sizes is to show that our task formulation is model-independent and to understand the performance improvement as the model size scales in different learning setups.

#### 6.3.1 In-Domain Workflow Actions

Table 6 shows the results of our methods trained on all ABCD domains and tested on the ABCD test set. We report each model variant's Cascading Evaluation (CE) and Exact Match (EM). Further, to understand the added complexity of predicting the slot values, we report the Action-only CE and Action-only EM results, where we are only interested in the models' ability to predict the actions correctly regardless of the predicted slot values.

Table 6: WD in-domain test results on ABCD. With and Without Possible Actions represents the results with and without the possible actions added to the input. EM and CE are the Exact Match and Cascading Evaluation, respectively. Action-only EM and CE are the EM and CE scores for Action-only prediction, excluding slot values.

| Params | Models | EM/CE | | Action-only EM/CE | |
|--------|--------|---------------------------------|------------------------------|---------------------------------|------------------------------|
| | | Without Possible Actions | With Possible Actions | Without Possible Actions | With Possible Actions |
| 60M | T5-Small | 44.1/67.9 | 44.8/68.6 | 55.4/78.6 | 56.7/79.1 |
| 220M | T5-Base | 47.7/69.9 | 49.5/72.3 | 56.2/79.4 | 57.5/79.2 |
| 406M | BART-Large | 42.0/60.3 | 44.9/64.3 | **61.9**/68.8 | **64.3**/70.1 |
| 568M | PEGASUS-Large | 49.9/71.2 | 52.1/72.6 | 59.6/81.2 | 62.9/82.8 |
| 770M | T5-Large | **50.6/73.1** | **55.7/75.8** | 59.9/**81.4** | 63.3/**83.1** |

With and without adding the possible actions, the EM and CE scores show an expected improvement as we scale the model size except for the BART-Large variant. Our analysis showed that the performance gains as we scale the model size are due to a better ability to extract slot values. The difficulty of predicting the slot values is because ABCD contains 30 unique actions with 231 associated slots, with each action having either 0, 1, or 3 slots. Further, some actions use different slots depending on the customer's answer. For example, "*pull up customer account*" can use either the account ID or the customer's full name. Our qualitative analysis showed that 37% of the slot values failures arise when both the "*pull up customer account*" and "*verify-identity*" occur in the same dialogue. In such a situation, a customer provides the account ID and the full name (present in 32.5% of test dialogues), and "*pull up customer account*" can use either slot and "*verify*

*customer identity*" uses both. Using the account ID or the full name for "*pull up customer account*" is valid from the ABCD's agent guidelines perspective. However, it might not match the gold truth. Further, larger models perform better on dialogues with turn counts larger than 20, which represents 38.2% of test dialogues. This improvement is related to large models' ability to handle longer context (Tay et al., 2021; Ainslie et al., 2020). As for BART-Large variant, our qualitative analysis showed that it exhibits an interesting behavior where it performs poorly on extracting some slot values that represent either names, usernames, order IDs, or emails. For example, it predicts "*crystm561*" instead of "*crystalm561*", knowing that the former does not exist in the dataset. The fact that 70% of the test workflows contain slots of this type explains the drop in performance. This observation is also confirmed by the high Action-only EM score, where slot value predictions are ignored. Our analysis showed that BART-Large performs better on target workflow with at most two actions. However, we did not find a clear explanation for this behavior. Further, while T5-Large shows an improvement over the smallest model (i.e., T5-Small), the EM score is only slightly above 50%, which shows the difficulty of the WD task. Moreover, the significant difference between the EM and CE scores across all configurations shows that it is much harder to predict the actions and their slot values in the exact order, which shows the utility of CE as a good metric for understanding the performance on the WD task.

Adding the possible actions improves the performance of all model variants, especially larger ones. For example, T5-Large improved by 5.1 on the EM score suggesting that the results should improve with even larger models (i.e., more than 770M parameters). However, the T5-Small variant improved by only 0.7 on the EM score. We believe that it is related to the fact that smaller models have difficulty utilizing larger inputs, a known limitation of Transformer models (Tay et al., 2021; Ainslie et al., 2020). Our qualitative analysis showed that most of the improvements gained by adding the possible actions are due to better slot value prediction, which can be counterintuitive. However, we believe that the models use more capacity to extract the slot values since providing the actions makes the action prediction an easier task. Further, adding the possible actions improved the action prediction performance on dialogues with turn counts larger than 25 (16.2% of test dialogues), which is shown by the higher Action-only EM and CE scores.

To better understand the performance of our proposed baselines, we compared our results to human performance on 100 randomly selected test samples representing 10% of the ABCD test set. We hired two annotators to label the samples and provided them with the complete list of actions with descriptions extracted from the ABCD agent's guidelines. Moreover, we provided three fully annotated dialogues for each action from the ABCD training set. However, the Cohen kappa coefficient was 0.63, indicating a low agreement between the annotators. All the disagreed items were labeled by a third annotator[5]. Table 7 shows the EM and CE scores on the annotated subset.

Table 7: Human evaluation results on ABCD test subset. EM and CE are the Exact Match and Cascading Evaluation, respectively.

| Methods | EM/CE |
|---|---|
| Human Performance | 11.0/38.12 |
| T5-Small Without Possible Actions | **42.3/66.0** |

The results show that even our smallest model outperforms human performance by a large margin. Our analysis showed that our method outperforms human annotators in cases where the agent performs multiple actions at the same turn. Further, our approach outperformed human performance in dialogues with more than 13 turns. However, since ABCD has 30 domains with 231 associated slots, further annotator training could be beneficial since the annotators reported difficulties when annotating the dialogues shown by the low Cohen kappa coefficient.

We also ran an ablation study to investigate the positive effect of shuffling and randomly selecting the actions added to the input described in Section 4.1. The result of this study is shown in Appendix A.1.1, but in summary, this technique improves the performance of all model variants and avoids a performance drop for our small model (i.e., T5-Small).

---

[5]The annotated human subset is available in our public code repository.

### 6.3.2 ABCD Cross-Domain Zero-Shot

We performed a "leave-one-out" cross-domain zero-shot experiment similar to Lin et al. (2021b); Hu et al. (2022); Zhao et al. (2022b) on ABCD. In this setup, we train a model on samples from all the domains except one. Then, we evaluate the performance on the test set that includes the omitted domain. In this experiment, we excluded the "Shirt FAQ" and "Promo Code" domains by removing all training samples that contain actions unique to each domain. The "Shirt FAQ" domain consists of 34 dialogues representing 3.4% of the ABCD test set. It overlaps other domains, such as "Boots FAQ" and "Jacket FAQ" on two actions (i.e., "*search faq*" and "*select topic in faq*"). However, while other domains use the "*select topic in faq*" action, the "Shirt FAQ" domain has eight unique slot values for this action since they represent existing FAQ questions that agents have to choose from (described in ABCD's agent guidelines). The "Promo Code" domain consists of 53 dialogues representing 5.3% of the ABCD test set. It overlaps other domains on three actions (i.e., "*pull up customer account*", "*ask oracle*", and "*check membership level*"). However, the overlapped actions appear in an order unique to this domain, making it more difficult since the order is important in the WD task. Table 8 shows the results of both experiments.

Table 8: WD leave-one-out cross-domain zero-shot results on ABCD with "Shirt FAQ" and "Promo Code" as target domains. With and Without Possible Actions represents the results with and without the possible actions added to the input. EM and CE are the Exact Match and Cascading Evaluation, respectively.

| | | Promo Code EM/CE | | Shirt FAQ EM/CE | |
| | | Without Possible Actions | With Possible Actions | Without Possible Actions | With Possible Actions |
| Params | Models | | | | |
|---|---|---|---|---|---|
| 60M | T5-Small | 42.3/65.1 | 42.5/66.5 | 41.0/64.8 | 41.0/65.0 |
| 220M | T5-Base | 46.0/67.6 | 47.8/69.7 | 45.1/68.8 | 45.7/69.0 |
| 406M | BART-Large | 41.5/58.3 | 43.6/62.2 | 40.0/59.1 | 42.3/61.7 |
| 568M | PEGASUS-Large | 47.4/68.8 | 49.6/69.2 | 46.2/68.9 | 47.4/69.3 |
| 770M | T5-Large | 48.1/70.7 | **51.8/72.3** | 49.9/72.4 | **50.2/72.4** |

The results show that our approach achieves good zero-shot transfer performance in both domains. Overall, the behavior of the different variants on both domains with and without the possible actions follows the same trends as the in-domain results in Table 6. Similarly, BART-Large performs poorly on both domains, and our analysis showed that it suffers from the same issues described in Section 6.3.2. However, BART-Large has the least amount of performance drop compared to the in-domain results since the target domains do not contain the slots in which this model performs poorly, as described in Section 6.3.2. On average, the performance on the "Shirt FAQ" is higher than the "Promo Code" domain, where the latter has an EM score 3.5 points lower than the in-domain results, showing that the "Promo Code" domain presents a more difficult cross-domain zero-shot setup.

Our results analysis showed that on the "Promo Code" domain, all the failures relate to the "*offer promo code*" action. When the possible actions are not used, most of the failures when using T5- were caused by the model copying parts of the input. However, for larger models, 67% of the failures are caused by the models predicting "*offer refund*" instead of "*offer promo code*". However, it is worth noting that dialogues from the "Offer Refund" domain unfold similarly to the "Promo Code" domain. In such situations, a customer is unhappy with a purchase, and the agent offers a refund, similar to offering a promo code. Further, the slot value (i.e., the promo code) is correctly predicted in 80% of these cases (i.e., when predicting "*offer refund*"). We believe that this is caused by the fact that the slot value of "*offer refund*" (.i.e., the refund amount) is close to the one of "*offer promo code*". However, this does not show in the results since both metrics require both the action and slot values to be correct. In the remaining, 33% of the failures, the larger models generated plausible action names such as "*generate new code*", "*create new code*", and "*create code*" (See Appendix A.3 for more details plausible action names). These predictions were not deemed valid since their BERTscore was lower than the threshold. On the other hand, larger models predicted actions that are considered valid such as "*offer code*", "*offer the code*", and "*provide code*". When the possible actions are used, the failures due to the "*offer refund*" confusion are reduced to 33%, showing the efficacy of adding the possible actions

to the input. The remaining errors were due to either invalid slot value prediction or invalid generated action. In this setup, 70% of valid predictions used the exact action name provided in the input, and the remaining were generated actions that are considered acceptable. On the "Shirt FAQ" domain, When the possible actions are not used, 71% of the failures are due to confusion with similar actions such as "search for jackets", or "search for jeans". The remaining errors were due to bad slot value prediction. Further, larger models generated plausible actions (e.g., "*search t-shirt*" or "*information on shirts*" and slot values (e.g., "does this *t-shirt* shrink" instead of "does this *shirt* shrink"). See Appendix A.3 for more examples. When the possible actions are added to the input, only 15% of the failures were due to confusion with similar domains (mostly "Jacket FAQ"), and the remaining failures were due to invalid slot value predictions. In this setup, 83% of valid predictions used the exact action name provided in the input, and the remaining were generated actions that are considered acceptable.

We also performed an ablation study on the randomization technique and showed that it has a similar effect to the one seen in the in-domain setting. See Appendix A.2 for more details.

### 6.3.3 MultiWOZ Cross-Dataset Zero-Shot

In this experiment, we evaluate if a model trained on one dataset can be used on another from a different domain in a zero-shot setup similar to Zhao et al. (2022b). Therefore, we test our models trained on ABCD described in Section 6.3.2 on all MultiWOZ test set domains. In this setting, there is no inter-dataset overlap. All the eight MultiWOZ domains are different from the ones in ABCD. We report the results of this experiment in Table 9.

Table 9: WD cross-dataset zero-shot test results on MultiWOZ. With and Without Possible Actions represents the results with and without the possible actions added to the input. EM and CE are the Exact Match and Cascading Evaluation, respectively.

| | | EM/CE | |
|---|---|---|---|
| **Params** | **Models** | **Without Possible Actions** | **With Possible Actions** |
| 60M | T5-Small | 0.0/0.0 | **5.3/8.4** |
| 220M | T5-Base | 3.6/10.0 | **23.0/38.3** |
| 406M | BART-Large | 5.1/12.2 | **24.1/39.9** |
| 568M | PEGASUS-Large | 9.8/15.2 | **27.4/41.2** |
| 770M | T5-Large | 9.0/13.1 | **26.9/40.0** |

As the results show, all model variants performed poorly when the list of actions was not included in the input. For example, T5-Small was not able to predict any valid workflow and our largest model achieved only a 9.0 EM score. For the T5-Small model, Our analysis showed that 91% of the failures are cases where the model uses ABCD's actions, and the remaining are caused by copying the input. For all other models, 87% of the failures are due to the models using ABCD's actions on average. The remaining failures are cases where the models copy part of the input or invalid action or slot value predictions. In this setting, all valid predictions were for cases with a single slot value, and the generated action names were considered valid using our evaluation method. For example, the models generated "*searching for hotels*", "*Looking for hotel*", and "*get a taxi*". However, when adding the possible actions, the performance improved for all variants with an average EM score increase of 18.3 points, showing the utility of this technique in the zero-shot setting. In this setting, 65% of the valid predictions are exact matches to the one provided in the input, and the remaining are generated action names that are deemed valid. Our analysis showed that 80% of the failures are due to invalid slot value predictions, especially for actions with more than three slot values. We believe this is related to the fact that the source domains (i.e., ABCD) have at most three slot values per action. The remainder of the failures are due to using ABCD's action names. Further, PEGASUS-Large performs better than WD-T5-Large even if it has 200M fewer parameters. Our qualitative results analysis shows that WD-T5-Large uses actions from the source domains (i.e., ABCD) more often than WD-PEGASUS-Large. For example, it uses "*search-timing*" for cases where a customer asks about the train departure time. Moreover, BART-Large performs better in this setting since MultiWOZ does not contain usernames or email slot values. Furthermore, we noticed that some predicted slot values are more refined than the gold ones (e.g., *museum*

*of science* instead of *museum*). However, our evaluation method does not mark such predictions as valid. Further, while adding the possible actions improves the performance, it presents a limitation for cases where this list is unknown and prompts future work to find an unsupervised method to extract it. The same limitation applies to the slot values, where one can solve the performance drop similar to Zhao et al. (2022b) by providing the slot descriptions in the input. However, in some cases, this list is unknown or frequently updated.

In both with and without possible actions, models larger than T5-Small were able to generate plausible action names. For example, the models generated the following: "*search for a swimming pool*", "*update booking information*", "*search for a table*", "*make a reservation*", "*search for a museum*", "*search for guesthouse*", and "*search for a room*". While some of these predictions can be ambiguous (e.g., "*make a reservation*"), others are very plausible. For example, "*search for a room*" or "*search for guesthouse*" are valid predictions for "*search for hotel*", and "*search for a museum*" is a valid prediction for "*search for attractions*" since the museum is an attraction. While such predictions are not exact matches, they provide valuable domain knowledge, especially in a zero-shot setting. However, our evaluation scheme cannot mark these predictions as valid since they yield a BERTscore lower than the threshold. While lowering the threshold could solve this issue, it will increase the false positives since the action names are short sentences. For example, "*search for hote*" and "*buy hotel*" has BERTScore of 89%, which shows one of the limitations of our proposed evaluation method.

Finally, it is worth noting that in the zero-shot setting, when the list of actions is not provided, the models trained on WD perform a form of intent (or action) and slot induction. Here the models generate the action names and extract the matching slot values rather than just clustering utterances (Hu et al., 2022) and performing span extraction for slot values (Yu et al., 2022). However, one limitation of using WD as a way to perform intent and slot induction is that using WD on multiple dialogues could yield action names that are semantically similar but use a different syntax (e.g., "*searching for hotels*" and "*Looking for hotel*"). One way of solving such an issue is to use a similarity measure to group the generated actions (or intent) and choosing one of them as the cluster label in a post-processing step. One could also feed the chosen action names to the model when performing the WD task, increasing performance as shown previously. However, we leave this exploration for future work.

### 6.3.4 MultiWOZ Cross-Dataset Few-Shot

We consider the case where only a few samples are available for each workflow action. To this end, we conducted a cross-dataset few-shot experiment, where we trained a model on all ABCD domains, then fine-tuned it with randomly selected $k$ samples per action from all MultiWOZ domains. We picked $k = 1$, $k = 5$, and $k = 10$ that yielded a training set of 11, 55, and 106 samples, respectively. Due to the nature of workflows containing multiple actions, some of these actions might have more than $k$ samples.

Table 10: WD cross-dataset few-shot test results on MultiWOZ. With and Without Possible Actions represents the results with and without the possible actions added to the input. EM and CE are the Exact Match and Cascading Evaluation, respectively.

| $k$ | # Samples | EM/CE | |
| --- | --- | --- | --- |
| | | **Without Possible Actions** | **With Possible Actions** |
| 1 | 11 | 5.9/54.8 | 8.2/58.7 |
| 5 | 55 | 24.4/65.4 | 61.7/84.4 |
| 10 | 106 | 43.2/73.0 | **72.2/89.1** |

We report the few-shot results of our WD-T5-Base in Table 10. Our approach shows a significant adaptation performance that keeps increasing as the number of training samples increases, reaching an EM score of 72.2. Moreover, adding the possible actions to the input shows an important performance gain in all the settings, where it improved the EM score by an average of 32.8 points. Our analysis showed that the results follow the same behavior on the cross-dataset zero-shot experiment with $k = 1$. Then, all other values of $k$ follow similar behaviors as in the ABCD in-domain experiment. Similar to the cross-dataset zero-shot experiment, the failures are mainly due to invalid slot values prediction. However, when $k = 10$ the slot values issues

reduce considerably. Furthermore, the model generated 23 invalid actions as opposed to 5 when the list of actions is provided in the input. This behavior shows the utility of this conditioning mechanism and proves that it can restrict the model to the provided actions.

### 6.3.5 Using Natural Language Action Names

In ABCD, the action names are given in a label-like format (e.g., "pull-up-account"). However, this format could have a negative effect on the WD performance when using pre-trained models that might not have seen such a format during training since they are usually trained on well-written, well-structured text (Raffel et al., 2020a; Karpukhin et al., 2020). To validate this claim, we manually generated natural language (NL) action names for each action. For example, we choose "pull up customer account" as the NL action name for "pull-up-account" (See Appendix A.4.2 for the full list). Then, we run an experiment similar to 6.3.1 to compare the performance of all the model variants using either the original label-like names (e.g., "pull-up-account") or the NL action names (e.g., "pull up customer account"). For this experiment, we only compare the Action-only EM and Action-only CE since we are interested in the performance of correctly generating the action names and not the slot values. Table 11 shows the results of this experiment.

Table 11: Results of using original label-like names compared to natural language action names. Action-only EM/CE are the EM and CE scores for action-only prediction, excluding slot values.

| Params | Models | Action-only EM/CE | |
| --- | --- | --- | --- |
| | | With Original Labels | With NL Names |
| 60M | T5-Small | 53.2/74.1 | **55.4/78.6** |
| 220M | T5-Base | 52.8/74.1 | **56.2/79.4** |
| 406M | BART-Large | 59.2/66.3 | **61.9/68.8** |
| 568M | PEGASUS-Large | 56.7/76.5 | **59.6/81.2** |
| 770M | T5-Large | 56.6/85.9 | **59.9/81.4** |

We observe that using NL names yields better performance on both metrics on all model variants, where it increased the Action-only EM by an average of 2.9. Our qualitative analysis showed that when using the original labels, most of the failures compared to using NL names are related to incorrectly generating the hyphen in the label. For example, the models generate "pullup-account", or "pull-up account" instead of the target "pull-up-account". Further, this behavior is more prominent for labels that consist of a single word (i.e, "instructions" and "membership") since they represent only 7% of the labels, where the models generate invalid labels like "instructions - " or "-membership". Moreover, 60% of the errors made on these single-word labels are for the "membership" label. We believe this is due to the existence of a similarly named label (i.e., "search-membership"). Some invalid predictions include stop words (e.g., "pull-up-the-account"). While extending training could resolve these issues, it will negatively affect domain adaptation due to overfitting. Also, while some of these invalid predictions can be handled in a post-processing step, it will require additional effort to evaluate all the possible failures which can be a lengthy task, especially when adapting to a new domain. When using NL action names, the models still predict names that are not an exact match to the target. For example, the models predict "pull up the customer account" instead of "pull up customer account", or "ask customer try again" instead of "ask customer to try again". However, using NL names enables us to use techniques like stemming and ignoring stop words, avoiding the need to create elaborate post-processing procedures. However, stemming and ignoring stop words were used at most on 23% of the predictions (for T5-Small). The remaining improvements compared to using the original labels resulted from cases where the NL names brought semantic richness. For example, using NL action names, the models performed better on differentiating "check membership level" from "get memberships information" as opposed to using the original labels where the models confuse "search-membership" and "membership". However, manually generating the action names can be a laborious task when dealing with a large or frequently updated domain, and we leave exploring more automated ways to generate these names for future work.

### 6.3.6 Using Models Fine-tuned on Summarization

To understand the effect of using models fine-tuned on summarization, we performed an experiment following the same setup as in Section 6.3.1 without adding the possible action names to the input, and we compare the performance using the same model with and without summarization fine-tuning. We report the results of this experiment in Table 12 on all model variants except PEGASUS-Large since it does not have a pre-trained model that has not been trained on summarization tasks. Further, all models have been fine-tuned on CNN/DailyMail (Hermann et al., 2015) dataset.

Table 12: Results of using original label-like names compared to natural language action names. With and Without Summarization Fine-tuning represents the results with and without the use of models fine-tuned on summarization task. EM and CE are the Exact Match and Cascading Evaluation, respectively.

| | | EM/CE | |
|---|---|---|---|
| Params | Models | Without Summarization Fine-tuning | With Summarization Fine-tuning |
| 60M | T5-Small | 40.4/63.1 | **44.1/67.9** |
| 220M | T5-Base | 44.2/65.9 | **47.7/69.9** |
| 406M | BART-Large | **42.0**/60.2 | **42.0/60.3** |
| 770M | T5-Large | 50.5/**73.1** | **50.6/73.1** |

The results show that using models previously fine-tuned on summarization tasks improves the performance of models below 220 million parameters (i.e., T5-Small, and T5-Base). However, it did not have any effect on larger models' performance. Our result analysis showed that using fine-tuned models mostly improved the performance on slot value extraction for smaller models, especially for slot values with multiple words like full names (e.g., "joseph banter"). Further, the same performance was attained for larger models with 4.8% fewer training iterations when using summarization checkpoints. These results show that using summarization fine-tuned models for the WD task is beneficial.

## 7 Conclusion

We have formulated a new problem called Workflow Discovery, in which we aim to extract workflows from dialogues consisting of sequences of actions and slot values representing the steps taken throughout a dialogue to achieve a specific goal. We have examined models capable of performing three different tasks: Workflow Discovery, Action State Tracking, and Cascading Dialogue Success. Our experiments show that our sequence-to-sequence approach can significantly outperform existing methods that use encoder-only language models for the AST and CDS tasks, achieving state-of-the-art results. Furthermore, our method is able to correctly predict both in-domain and out-of-distribution workflow actions, and our action conditioning approach affords much better performance for out-of-domain few-shot and zero-shot learning. We hope that our work sparks further NLP research applied to the field of process mining, but allowing for the use logs, meta-data and dialogues as input. Moreover, we believe that this method of characterizing complex interactions between users and agents could have significant impact on the way researchers and developers create automated agents, viewing workflow discovery as a special type of summarization that produces a form of natural language based program trace for task oriented dialogues.

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

# A   Appendices

## A.1   Ablation on the Possible Actions Randomization

### A.1.1   In-Domain Setting

We run an ablation study using all our model variants to understand the effect of the randomization technique described in Section 4.1. The use of randomization is motivated by the fact that the number of possible actions can be different when adapting to a new domain or at test time. The result of this study is shown in Table 13.

Table 13: Randomization ablation results of in-domain WD on ABCD test set. w/ Possible Actions w/o randomization represents the results without randomization, and w/ Possible Actions the results with randomization as described in Section 4.1. EM and CE are the Exact the Match and Cascading Evaluation, respectively

| Model | EM/CE |
|---|---|
| WD-T5-Small | 44.1/67.9 |
| w/ Possible Actions w/o randomization | 42.6/66.9 |
| w/ Possible Actions | 44.8/68.6 |
| WD-T5-Base | 47.7/69.9 |
| w/ Possible Actions w/o randomization | 48.1/70.1 |
| w/ Possible Actions | 49.5/72.3 |
| WD-BART-Large | 42.0/60.3 |
| w/ Possible Actions w/o randomization | 43.2/61.0 |
| w/ Possible Actions | 44.9/64.3 |
| WD-PEGASUS-Large | 49.9/71.2 |
| w/ Possible Actions w/o randomization | 51.9/72.0 |
| w/ Possible Actions | 52.1/72.6 |
| WD-T5-Large | 50.6/73.1 |
| w/ Possible Actions w/o randomization | 54.6/74.8 |
| w/ Possible Actions | **55.7/75.8** |

The results show that adding the possible actions without randomization (i.e., w/ Possible Actions w/o randomization) improves the performance of all model variants except WD-T5-Small, where the EM score drops by 1.5 points. One explanation is that this model reaches its capacity since adding the possible actions lengthens the model input, which increases the complexity and reduces the performance, a known limitation of Transformer models (Tay et al., 2021; Ainslie et al., 2020). Nonetheless, our randomization technique (i.e., w/ Possible Actions) resolves this issue and improves, even more, the performance of all other model variants, especially larger ones. For example, WD-T5-Large EM score increased by 5.1 points. The performance increase of WD-T5-Small is due to the fact that input length is reduced at training time.

## A.2 Cross-Domain Zero-Shot Setting

Similar to the in-domain experimental setup, we performed an ablation study of the randomization technique described in Section 4.1 in the cross-domain zero-shot setting. Table 14 shows the result of this study.

Table 14: Randomization ablation results of cross-domain zero-shot WD on ABCD test set with "Shirt FAQ" and "Promo Code" as target domains. w/ Possible Actions w/o randomization represents the results without randomization, and w/ Possible Actions the results with randomization as described in Section 4.1. EM and CE are the Exact Match and Cascading Evaluation metrics, respectively.

| Model | Shirt FAQ EM/CE | Promo Code EM/CE |
|---|---|---|
| WD-T5-Small | 42.3/65.1 | 41.0/64.8 |
| w/ Possible Actions w/o randomization | 41.2/64.4 | 40.9/64.5 |
| w/ Possible Actions | 42.5/66.5 | 41.0/65.0 |
| WD-T5-Base | 46.0/67.6 | 45.1/68.8 |
| w/ Possible Actions | 46.9/68.7 | 45.3/68.8 |
| w/ Possible Actions w/o randomization | **47.8/69.7** | **45.7/69.0** |

Adding the possible actions (i.e., w/ Possible Actions) to the input has a similar negative effect on the smaller model (i.e., WD-T5-Small) as observed in the in-domain experiment results in Table 6, and similarly, it increased the performance of the larger variant (i.e., WD-T5-Base). However, our randomized actions

conditioning technique (i.e., w/ Possible Actions$^+$) resolves this issue and improves performance on both model variants.

### A.3 Plausible Predictions in the Zero-Shot Setting

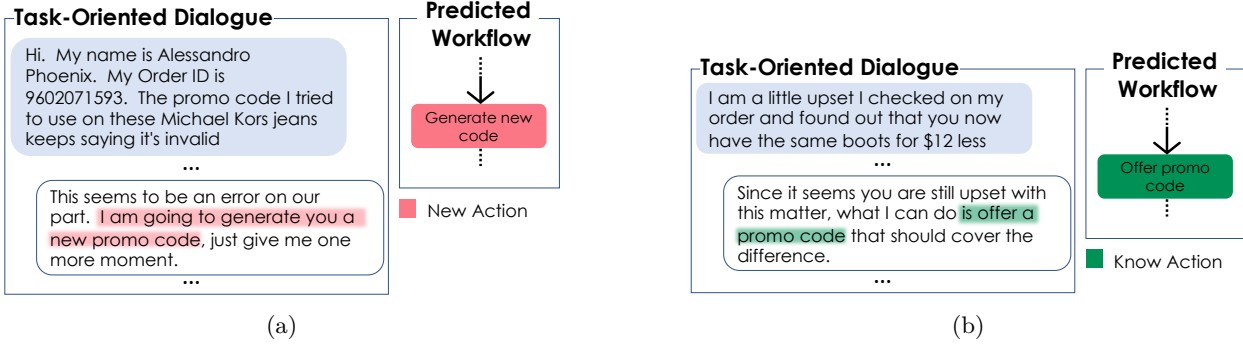

Figure 3: Plausible (a) and known (b) workflow actions generated during the ABCD cross-domain zero-shot experiment.

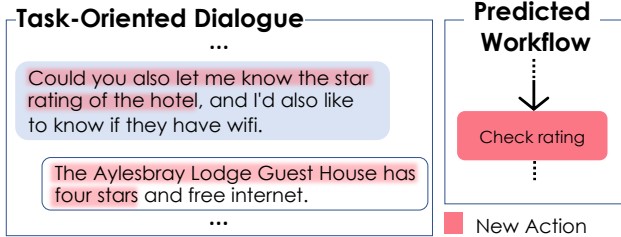

Figure 4: Plausible workflow action generated during the MultiWOZ cross dataset zero-shot experiment.

This section shows cases of plausible workflow actions predicted in the zero-shot setting that are not part of the possible actions. The generated actions are either "fine-grained" versions of existing actions or entirely new ones. For example, Figure 3a shows a case where the model predicted *generate new code* while the closet possible action is *offer promo code*. However, our analysis showed that there is a clear separation between cases in which agents offer a promo code as shown in Figure 3b and when they *generate* a new one if the old promo code is no longer working, as shown in Figure 3a.

Another example is shown in Figure 4 of an entirely new action that has no similarities with any other possible action. Here the customer is clearly asking for the rating of the hotel Hence the *check rating* action. Our dataset analysis showed that there are 10 test samples where customers requested to get the rating of hotels or restaurants.

### A.4 WD Dataset

### A.4.1 MultiWOZ Workflow Actions

Table 15 shows the MultiWOZ workflow natural language action names we used in all our experiments.

### A.4.2 ABCD Workflow Actions

Table 16 shows the ABCD workflow natural language actions names we used in all our experiments.

Table 15: MultiWOZ workflow actions

| Action Name | Natural Language Action Name |
| --- | --- |
| find_hotel | search for hotel |
| book_hotel | book hotel |
| find_train | search for train |
| book_train | book train ticket |
| find_attraction | search for attractions |
| find_restaurant | search for restaurants |
| book_restaurant | book table at restaurant |
| find_hospital | search for hospital |
| book_taxi | book taxi |
| find_taxi | search for taxi |
| find_bus | search for bus |
| find_police | search for police station |

Table 16: ABCD workflow actions

| Action Name | Natural Language Action Name |
| --- | --- |
| pull-up-account | pull up customer account |
| enter-details | enter details |
| verify-identity | verify customer identity |
| make-password | create new password |
| search-timing | search timing |
| search-policy | check policy |
| validate-purchase | validate purchase |
| search-faq | search faq |
| membership | check membership level |
| search-boots | search for boots |
| try-again | ask customer to try again |
| ask-the-oracle | ask oracle |
| update-order | update order information |
| promo-code | offer promo code |
| update-account | update customer account |
| search-membership | get memberships information |
| make-purchase | make purchase |
| offer-refund | offer refund |
| notify-team | notify team |
| record-reason | record reason |
| search-jeans | search for jeans |
| shipping-status | get shipping status |
| search-shirt | search for shirt |
| instructions | provide instructions |
| search-jacket | search for jacket |
| log-out-in | ask customer to log out log in |
| select-faq | select topic in faq |
| subscription-status | get subscription status |
| send-link | send link to customer |
| search-pricing | check pricing |

### A.4.3  Dataset Statistics

This section shows the dataset set statistics for the WD task for ABCD and MultiWoz datasets. Table 17 shows the size for each split, the average workflow length across each dataset, and the number of domains. Figure 5 and Figure 7 show the workflow length distribution for each data split on ABCD and MultiWoz, respectively. Figure 6 and Figure 8 show the action distribution for each data split on ABCD and MultiWoz, respectively.

Table 17:  WD dataset statistics for ABCD and MultiWoz datasets.

|  | # Train Samples | # Dev Samples | # Test Samples | Average Workflow Length | # Domains |
|---|---|---|---|---|---|
| **ABCD** | 8034 | 1004 | 1004 | 18.5 | 30 |
| **MultiWoz** | 5048 | 527 | 544 | 3 | 8 |

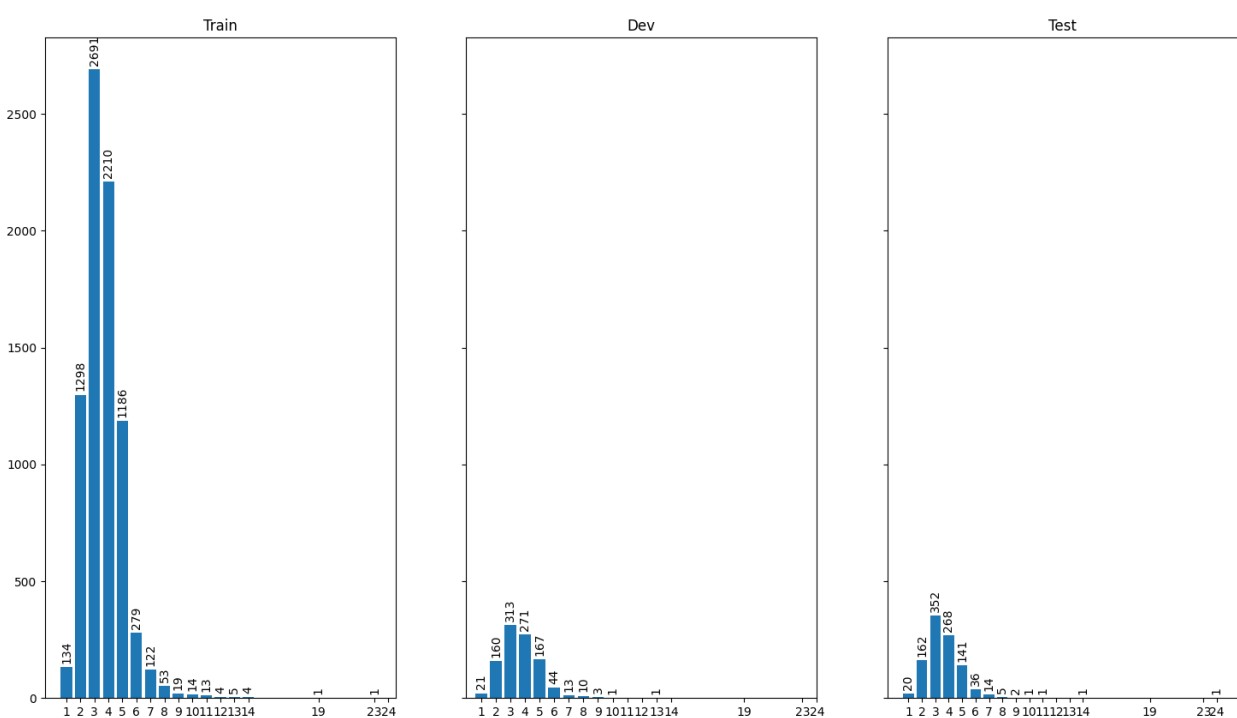

Figure 5: Workflow length distribution for each ABCD dataset split.

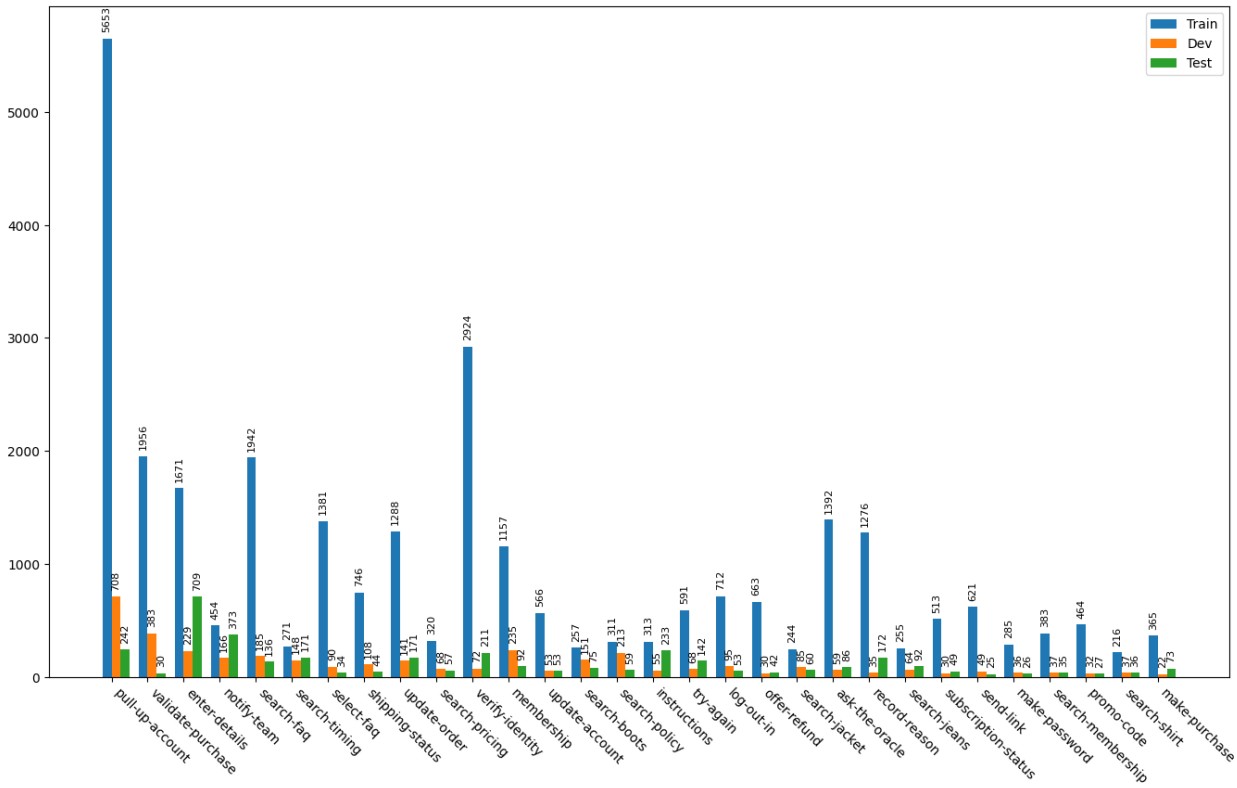

Figure 6: Action distribution for each ABCD dataset split.

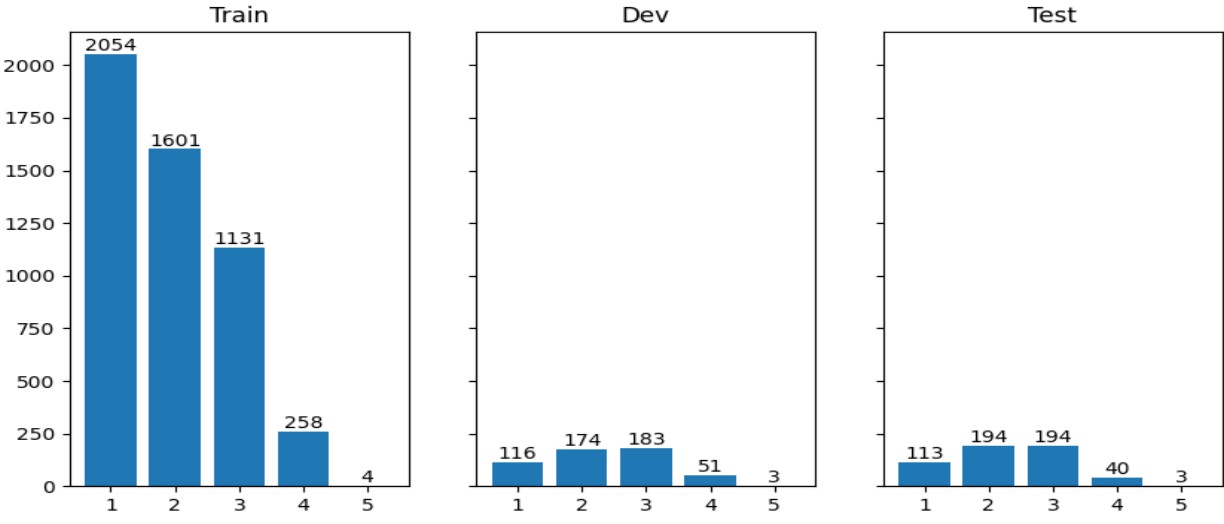

Figure 7: Workflow length distribution for each MultiWoz dataset split.

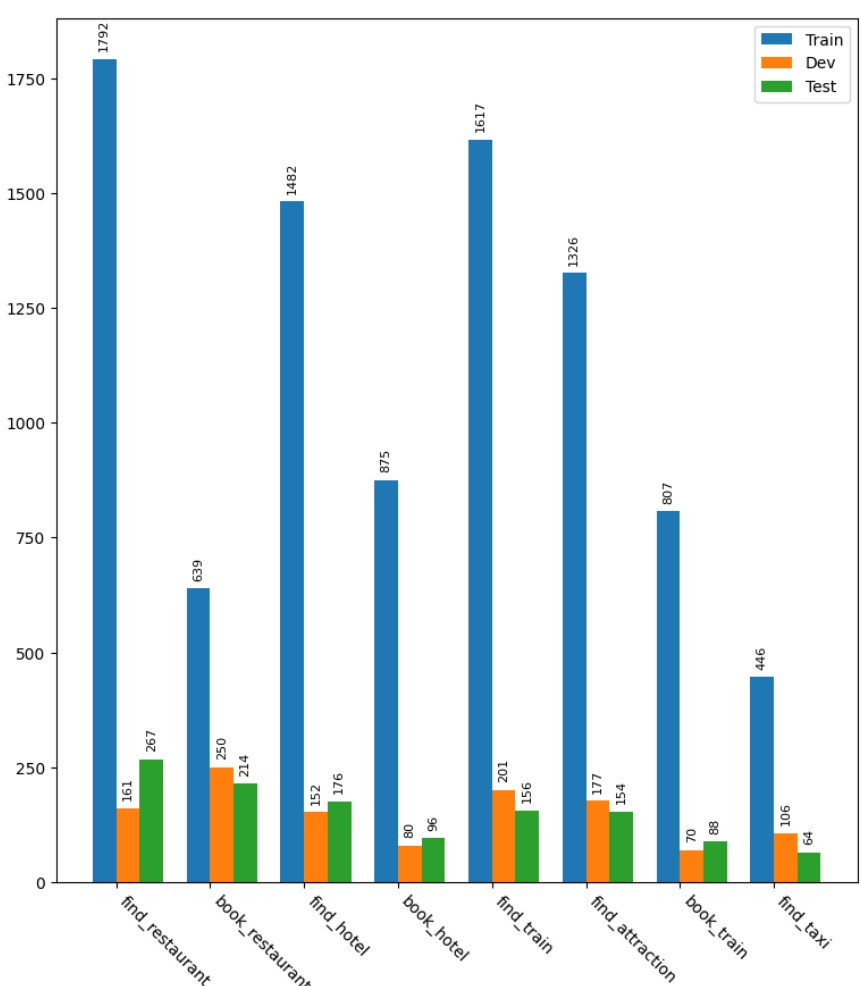

Figure 8: Intent distribution for each MultiWoz dataset split.

## A.5 Prompt and target formats for WD, AST, and CDS tasks

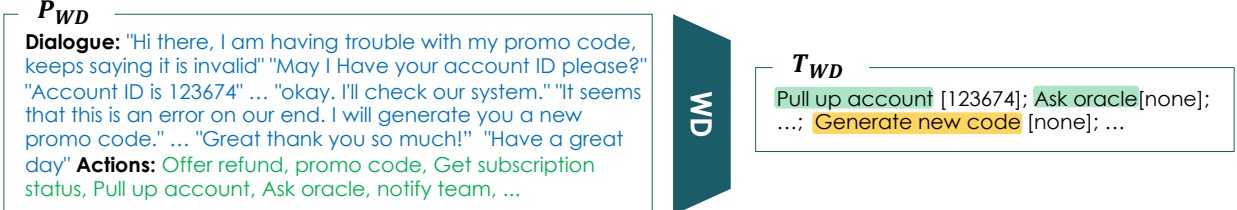

(a) **Workflow Discovery (WD)**. Illustration of a prompt ($P_{WD}$) – consisting of an entire dialogue, and the target ($T_{WD}$) – consisting of a structured summary for the WD task. $P_{WD}$ is composed of the delimiters in bold that separate the dialogue utterances in blue and the optional list of possible actions in green. $T_{WD}$ is composed of the predicted workflow actions, and matching slot values in brackets. The actions are either known (highlighted in green) or invented (highlighted in orange).

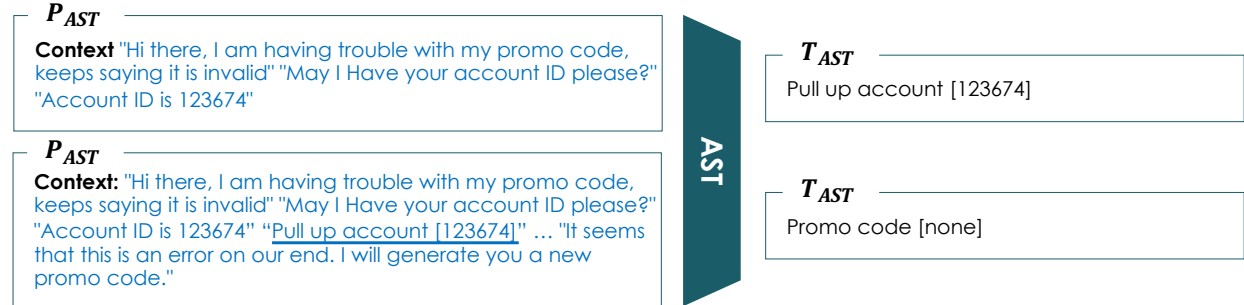

(b) **Action State Tracking (AST)**. Illustration of prompts ($P_{AST}$) and targets ($T_{AST}$) for the AST task at different turns in the dialogue. $P_{AST}$ is composed of the context delimiter in bold and the context utterances in blue that can contain previous action turn text (underlined).

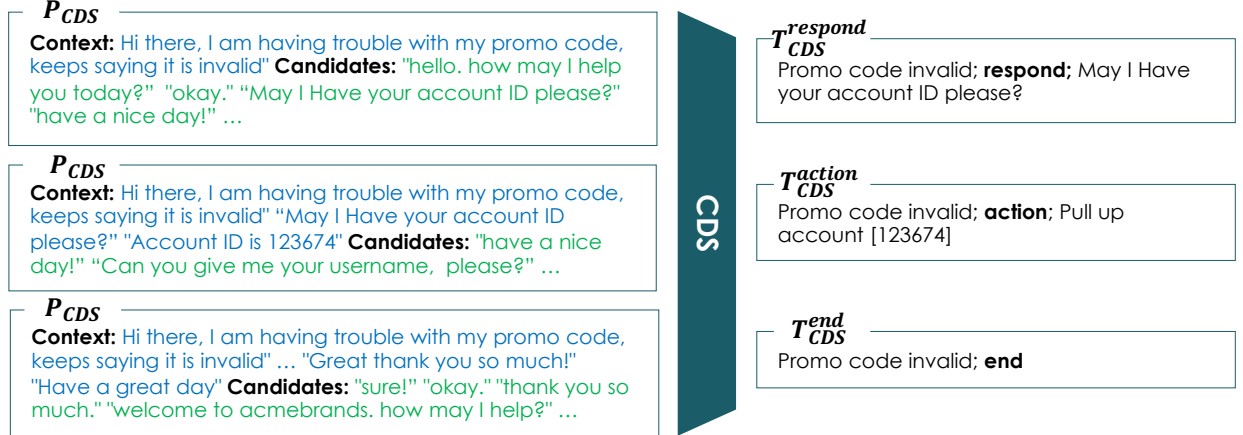

(c) **Cascading Dialogue Success (CDS)**. Illustration of prompts ($P_{CDS}$) and targets $T_{CDS}$ for the CDS task at different turns in the dialogue. $P_{CDS}$ is composed of the delimiters in bold that separate the dialogue utterances in blue and the list of possible agent response candidates in green. $T_{CDS}^{respond}$, $T_{CDS}^{action}$, and $T_{CDS}^{end}$ are the targets when the next step is to **respond** with an utterance, perform an **action**, and **end** the conversation, respectively. "Promo code invalid" is the entire dialogue intent.

Figure 9: Illustration of prompt and target formats for WD, AST, and CDS tasks.

