# OpenReview forum: "Workflow Discovery from Dialogues in the Low Data Regime"
_TMLR — Accepted by TMLR_

### Review · Reviewer_TjxB · 2022-11-14

**Summary Of Contributions:**

Introduce a task for Workflow Discovery (WD) that aims to extract the actions and associated slot-values from a dialogue. Authors evaluate their ability to complete this task on ABCD using various Seq2Seq models.

Authors also study cross-dataset workflow extraction in zero-shot and few-shot settings.  In particular, conditioning on the knowledge from ABCD transfers over to MultiWOZ

Finally, authors offer SOTA results on AST and CDS, the two tasks proposed within the ABCD paper.


**Audience:**

Yes

**Broader Impact Concerns:**

The models and topics do not touch on sensitive topics, so no issues with broader impact or ethical concerns.

**Claims And Evidence:**

Yes

**Requested Changes:**

Authors should consider how workflow discovery is related to intent induction, which is a known task within dialogue settings.  Intent induction or slot-induction is more similar to WD than summarization, so a mention should certainly be added to Related Works and discussed in other areas.  Some examples include:
  - https://aclanthology.org/E17-2078/
  - https://aclanthology.org/D19-1413/
  - https://arxiv.org/abs/2103.08886
  - https://aclanthology.org/2022.naacl-main.86/

Since the journal places less emphasis on novelty, the fact that the proposed WD task is similar to a known task (intent induction) is not a critical issue.  Mentioning the relationship somewhere is necessary, but should be straightforward enough to accomplish.  With that said, if it is possible to include an intent induction baseline for comparison to the text-to-text methods, that would significantly strengthen the work.

Clarify in Table 5 that EM*/CE* refer to actions only.  Perhaps in the description, but ideally somehow within the table.  Also, the table might benefit from being split, one to show different models (BART, Pegasus, T5) and one to show size (small, base, large).  Could use two columns, since there is enough space.

Move some of the ablations into the main section and add more analysis around having the list of actions vs. no list of actions.  This is much more impactful than randomly shuffling the list of actions.
Possibly adding some line graphs, rather than just tables, since it might be easier to understand the trends.

Typos
 - "trained on another domain is a few-shot setting" page 2
 - "In the MutliWOZ dataset" page 7

**Strengths And Weaknesses:**

Strengths
  - Clearly explains the proposed task (WD) and its relation to similar tasks.
  - Runs thorough experiments across a wide variety of model architectures, datasets, input setups and model sizes.
  - Strong state-of-the-art performance on the AST and CDS tasks for ABCD.

Weaknesses
  - Missing a comparison to intent and slot induction for dialogue.
  - Authors mention that Workflow Discovery task includes an optional list of possible actions.  Operating with a known list of actions vs. not knowing this list is an important distinction.  The latter is a much more realistic setup that would be of much greater interest.  Accordingly, the difference in the two settings should be analyzed more closely.
  - Analysis of techniques and ablations are limited.

---

> ### Author Response · Authors · 2022-12-08
> **Response for Reviewer TjxB**
>
> - **Missing a comparison to intent and slot induction for dialogue**: We have added a conceptual comparison to intent and slot induction in our literature review.
> - **Authors should consider how workflow discovery is related to intent induction, which is a known task within dialogue settings. Intent induction or slot-induction is more similar to WD than summarization, so a mention should certainly be added to Related Works and discussed in other areas**: We agree that intent induction is similar to WD and we have added an entire section for intent induction in the related work.
> - **If it is possible to include an intent induction baseline for comparison to the text-to-text methods, that would significantly strengthen the work.**: While we agree that adding this baseline can improve the work. However, to the best of our knowledge there is no prior work on intent induction that can automatically give meaningful names to new intents, in the manner that we can give new meaningful names to actions (due to our use of transformer techniques and prompting methods). Prior work is frequently based on clustering embeddings and clusters would need to be labeled. Clustering methods also tend to have ad-hoc parameters which need to be tuned. Since actions are also different from intents, prior methods would also need adaptation to our new WD scenario.
> - **Clarify in Table 5 that EM\*/CE\* refer to actions only. Perhaps in the description, but ideally somehow within the table. Also, the table might benefit from being split, one to show different models (BART, Pegasus, T5) and one to show size (small, base, large). Could use two columns, since there is enough space.**: We have reworked all the tables to utilize more space and explicitly changed EM*/CE* to Action-only EM/CE and added an explanation in the table descriptions and experimental setup.
> - **Move some of the ablations into the main section and add more analysis around having the list of actions vs. no list of actions. This is much more impactful than randomly shuffling the list of actions.**: We moved importance ablations to the main sections and moved the randomization ablation to the appendix. Further, we reworked all the experiment sections to include an in-depth result analysis.
> - **Possibly adding some line graphs, rather than just tables, since it might be easier to understand the trends.**: While we agree that graphs can make the trends easier to follow, when we tried creating the graph, we ended with a complex and hard-to-read since we have multiple models on 4 metrics (EM, CE, Action-only EM, and Action-only CE) with two setups for each (with possible actions and without possible actions). We believe that the rework we did to the tables makes them more readable. However, if you feel that a graph is still a better option for a specific case, we will be happy to do it.
> - **Typos**: We have fixed the typos, thank you for noticing.

---

### Review · Reviewer_dNS3 · 2022-11-22

**Summary Of Contributions:**

This paper formulates a problem of workflow discovery (WD) from dialogues. WD could be closer to a realistic setting where all possible actions (or their sequences) are not fully defined beforehand. For the explanation, WD is compared with other tasks, action state tracking (AST) and cascading dialogue success (CDS). Based on the intuition that the WD problem is similar to summarization, text-to-text models (T5, BART, and PEGASUS) are used, fine-tuned to three tasks (WD, AST, and CDS), and evaluated on both an in-domain scenario (ABCD) and cross-domain scenario (MultiWOZ) based on a conditioning mechanism.

**Audience:**

Yes

**Broader Impact Concerns:**

The experiments are mainly conducted on the ABCD dataset (and MultiWOZ). Whether the proposed method could be extended to general or more realistic datasets are questionable.

**Claims And Evidence:**

Yes

**Requested Changes:**

Please include important baseline results and human accuracy to understand how models from the proposed method perform well relatively and absolutely.

Please add more literature on sequence-to-sequence models and few-shot/zero-shot approaches for AST and CDS.


**Strengths And Weaknesses:**

The paper is well structured. The new problem formulation looks interesting. However, whether this task is really useful needs to be explained better. Also, the used models are quite straightforward for the WD task.

The claim that workflow discovery is similar to summarization is not intuitive enough. How much summarization models give gains to the WD task is also not demonstrated.

The superiority between T5, BART, and PEGASUS is ambiguous. This is also the case with summarization datasets. However, can you provide why similar situations occur for these tasks? Also, considering this, can you provide a solution about which model practitioners should use for new datasets?

I think the WD problem can be factorized into AST and CDS by dividing multiple turns. Do you have any thoughts on this? Also, there could be several variants of the problem formulation. Could you clarify why the proposed formulation is legit and most effective?

Formats of inputs, outputs, and prefixes for each task are manually designed. I wonder how the results could change depending on different formats.

The baseline might need to be stronger to say that the proposed method is the-state-of-the-art.

---

> ### Author Response · Authors · 2022-12-08
> **Response for Reviewer dNS3**
>
> - **Usefulness of WD needs to be explained better**: We added more detail on the usefulness of the proposed WD task in the introduction (paragraph 2). We also added information on how the WD task differs from existing tasks such as AST, DST, and Intent induction which shows, even more, the utility of WD. We believe that it would be quite useful to have a tool that reads dialogues between two humans and identify simple sequences of steps taken to resolve different types of problems during the discourse. Such an analysis could be a first step towards completely automating processes that are performed online through text interfaces or over telephone calls. The extracted sequence can also be used to detect cases where agents diverge from formal workflows prompting further agent training since it can affect customer satisfaction where task completion depends on the agent and not the task
> - **How much summarization models give gains to WD task**:  Interesting question. To understand the benefit of summarization we added a new experimental section showing the benefits of using summarization in section 6.3.6. In short, smaller models benefit more from summarization pre-training, and larger models train faster. The produced workflows can be seen as a summary of the agent's actions.
> - **Superiority between T5 BART PEGASUS ambiguous**: We reworked Table 6 to delve deeper into the properties and performance differences of these models. This new table includes the number of params because while these models are named Large, they are of different sizes. These and other factors help us understand the performance differences. We also included extensive analysis showing the different issues for each model in each learning setup
> - **model for new datasets**: the obvious choice is to choose the best-performing model depending on the learning setup. Practitioners might wish to account for the extra training complexity, memory footprint, and inference time due to larger models
> - **WD factorized into AST & CDS**: We added information to detail the difference between these tasks, including a table summarizing these differences (now Table 1). Some aspects of AST could be seen as a subset of WD when actions and slots are known. However, AST & CDS require annotated action turns which is unique to ABCD. Whereas WD operates in a setup where actions and slots are unknown, or not fully known. Moreover, when all actions and slot values are known, running AST after each action turn could generate a set of actions that might resemble a workflow. However, it will contain the set of actions following the agent guidelines (Chen et al 202). Whereas WD extracts the set of actions the agent performed regardless of the guidelines
> - **results change depending on different formats**: We included an experimental section that compared using label-like action names to natural language actions names and showed that the latter performs better, showing that the format affects the results, but this is a known fact of seq2seq models where the prompting format is by itself a hyperparameter (Raffel et al 2020). However, we did explore further since our main goal is to create a good baseline for the WD task and not find the best model
> - **important baseline & human accuracy**: We have added human accuracy and reworked all experiment sections to include more analysis. However, were only able to label 10% of the test set due to the rebuttal time constraint
> - **literature on seq2seq models & few/zero-shot AST & CDS**: We redid all related work to add seq2seq models. We didn't find prior work on few-shot/zero-shot AST & CDS. However, we added work on intent/slot induction which can be seen as the zero-shot version of AST.
> - **Extendability realistic datasets is questionable**: WD does not require action turns. So it can easily be extended to other datasets. Further, we can use our method in realistic use cases, because we don't require to know the exact turn when actions occur. One can create a dataset by extracting logs from systems used by agents and extract all logged actions. Then, we can train a model on WD using the actions (representing the workflow) and matching utterances. Also, our prompt does not include speaker labels making our method usable in situations where this information is unavailable or inaccurate (using ASR)
> - **SOTA claim**: First, we are not claiming a SOTA result on our WD task. SOTA is claimed on AST and CDS. However, since it does not outperform prior work on all metrics we will be more precise with our language and only refer to the metrics where our variant has the best-known result.
> - **why the proposed formulation is legit and most effective?**: Testing our text-to-text framework on CDS & AST tasks aimed to validate our formulation, and the strong results on AST and CDS confirm that. However, we don't claim that this formulation is the most effective since our goal is to provide a strong enough baseline for the proposed task.

---

### Review · Reviewer_ST5G · 2022-11-23

**Summary Of Contributions:**

This paper introduces a novel variant of the action state tracking task that they named Workflow Discovery.
In short, the agent needs to output the workflow $T_{WD}$, i.e., a sequence of actions (with its labels), given a dialogue and a set of actions.
Hence, WD may be seen as an action state tracking task where the action tags are provided in the input (while AST actions are defined beforehand as an output). This task formulation allows for more flexibility -- few/zero-hot learning -- and may naturally leverage pretrained large language models.

The authors first attempt to frame the WD task within the literature and compare it with other dialogue tagging tasks (AST, DST).
Then, they explain how to adapt the AST setting to the WD setting by defining a tokenization process for the input/output of a transformer.
Then, they evaluate their WD models in an AST setting, assessing that the model output reasonable results.
Finally, they assess diverse transformer architectures on different WD settings, e.g., in/across domain/dataset workflows, zero-shot, etc.


**Audience:**

Yes

**Broader Impact Concerns:**

Nan

**Claims And Evidence:**

Yes

**Requested Changes:**

The core changes I would recommend are the following:
 - Redo the full related work section with more focused on the AST/Cascade setting, with its strenghes, weaknesses, past attempts etc. Have a broader view on similar settings, e.g., convert dialog into SQL. There are many facets to explore.
 - clean the BibTeX - the formatting is inconsistent!
 - Introduce an abstract definition of WD, and then describe the tokenization/prompt system in an experimental section
 - remove the sota claim; it is simply out of the scope here
 - Improve and complete the experimental section: fuse and rework the experimental sections, add back bad results, rework the table to ease reading (fully type with/without action set), provide some qualitative analysis, and further experiment analysis - especially on the cross-dataset evaluation (which is the most exciting part in my opinion)
 - the randomized conditioning techniques ablation should be done once for all in an ablation section and should never be mentioned again (no need for the + every time)

(+ other see other remarks in previous paragraph)

**Strengths And Weaknesses:**

Strength:
 - First, the workflow task is an exciting direction that fits well with the spirit of recent NLP advances: more modularity in the task definitions. On a personal note, I think that it is an overstatement to call it workflow discovery, but it sounds nicer than sequential and conditional action tagging. I would encourage the authors to refine a bit the link between AST and WD to highlight better the core differences and the core similarities (e.g. a recap table). Overall, WD is more like a (welcome) generalization of AST.
 - Second, I appreciate that the authors assess their model on the AST tasks (which again demonstrate the close links between both methods) before testing them on the WD. It creates a proxy-baseline and gives trust in the next results.
 - Third, the paper flow is good. It is sometimes a bit wordy, but the reading is easy, and ideas are correctly articulated. It is easy to understand what the authors wanted to say.
 - The authors made some effort on the figures to ease intuitions, which is positive
 - The authors explore various WD settings with the inter/cross-domain and inter/cross dataset. As discussed below, those experiments may lack some in-depth analysis, but they go in the good directions

Weaknesses:
 - One of the biggest weaknesses is the bibliography work. For instance, the introduction does not contain any citations! Differently, dialogue system is among the oldest field in ML... but they are only three papers that were cited that are written before 2015. I do not require a full survey of pre-deep learning papers, but this clearly hints that the related work section is incomplete. When introducing a new task, the quality of the bibliography is of paramount importance. Here, it is incomplete, ill-dispatched and it does not credit past work (e.g., no citation when introducing  AST or CDS). For instance, this could be a good start: https://web.stanford.edu/~jurafsky/ws97/CL-dialog.pdf
 - The second weakness is that the paper confuses the WF definition and the experimental implementation. Indeed, they describe the task by introducing a prompting mechanism, which is transformer-dependent. The WF task could also be solved with a retrieval model. Therefore, I strongly recommend the authors to clearly differentiate the task, i.e. set of actions, each dialogue may have different action options, and the WF implementation, using a prompting mechanism.
 - The authors listed the dataset used to define WF. However, I would be in trouble reproducing the setting (split, final workflows, etc.). I would recommend the authors release a file containing their experimental setting. In addition, the authors should also provide some statistical analysis on the WF setting (average sequence length, action distributions, inter-domain action overlaps, inter-dataset action overlap, etc.)
 - The paper sometimes mixes task definition and model-proposal, which I found quite confusing. Indeed, while it is interesting to test the proposed model to the AST task, there is no point of claiming Sota here. I would strongly recommend stating that your model is good enough to define a strong baseline for the WF task. Similarly, introducing randomization ablation blurs the message: the point of the paper is not to look for a better model to solve WF, but to define the research problem
 - Finally, while I like the diversity of WF tasks, I think they are not explored enough. Only results are provided, and little analysis is performed, e.g., what happens with incomplete actions set, and how often a new/incorrect action was outputted. How do inter-dataset scores correlate with action overlap between dataset? What is the impact of turning the actions into words instead of new trainable tokens when using the same dataset? They are many ways to challenge the WF tasks, but none were performed


Overall, I would say that my core criticism of this paper is the lack of hindsight. While the authors introduce a task, there are too few references to the existing literature to rigorously frame their work in the community. Similarly, the author takes as granted the transformer prompt setting without abstracting the task by itself. Finally, the authors only compute final metrics and do not challenge their conclusion with additional experiments. While the paper has some potential, it needs more depth in the scientific background, task definition, and empirical analysis. However, the paper is good enough that a few rounds of iteration may solve these issues.

---

> ### Author Response · Authors · 2022-12-08
> **Response for Reviewer ST5G**
>
> - **introduction does not contain citations**: Thank you for pointing this out. We have reworked the introduction section to compare and contrast our WD framework with prior problem formulations, such as dialogue act modeling, intent and slot induction, DST and AST.
> - **refine link between AST and WD (e.g. a recap table)**: We added information about the difference between AST and WD and have added a recap table as suggested.
> - **Redo full related work with ... AST/CDS ... dialog to SQL**: We redid the related work to include the suggested subjects of AST/CDS and the similar setting of converting dialogue to sql. We added a new related work section on Intent and Slot induction, and structured text generation. We have also reworked the entire seq2seq related work.
> - **clearly differentiate the task, ... Introduce abstract definition of WD**: As requested, in section (3.1) we have added a formalized abstract task definition that is separate from our proposed, tokenization, architectures and prompting schemes for the problem.
> - **reproducing the setting ...release a file containing experimental setting**: We have added additional details to facilitate reproducibility, including for example the additional details in Section 5.2 on how we adapted ABCD for our WD experiments, and further details about the splits we used for our cross-domain zero shot experiments in 6.3.2. Our initial submission also contained supplementary material consisting of all the code used to generate the exact datasets we used. The source code also comes with a readme file that explains how to use it. However, we agree that we did not mention this in the body text of our work. We have correspondingly added better references to our code, and we will update it to be a public Github link after the double-blind review. We could include a separate text file with these details (i.e. extracted from sections 5.2 and others in the main text), or we could extend our readme to include experimental setup details if you think this would be preferred.
> - **statistical analysis on the WF setting**: We have added a new section in the appendix (APP A.4.3) with data statistics for both MultiWoz and ABCD. Further, we added inter-domain overlap in the cross-domain zero-shot experimental section (6.3.2) and cross-dataset information in the cross-dataset zero-shot experimental section (6.3.3).
> - **claiming Sota**: We really just wanted to examine the behavior and performance of one of our base architectures on the original ABCD AST task so that the reader has a point of reference. It turned out that our smallest T5 model outperformed prior work on multiple metrics. Since it does not outperform prior work on all metrics we will be more precise with our language and only refer to the metrics where our variant has the best-known result. In case it is not clear, we are not claiming a SOTA result on our own newly introduced WD problem and evaluation.
> - **randomization ablation blurs the message**: We agree that the randomization ablation adds confusion. So we have moved the randomization ablation to the appendix and reworked all the tables.
> - **Improve experimental section ... results not explored enough. ... challenge the WF none were performed**: We have added substantial additional analysis of our experiments across the Experimental Results section and added detailed empirical and qualitative analysis for all experimental setups, we added back bad results, and reworked all the tables as suggested. There is no inter-dataset action overlap between ABCD and MultiWOZ. We have added an explicit discussion of this in 6.3.3. We would contend that our zero-shot experiments are a pretty strong challenge to the WD tasks.
> - **impact of turning the actions into words ...?**:  While using trainable tokens might have a positive effect since we are reducing the decoding complexity, we did not explore such a method since we were interested in performing zero-shot learning, where the semantic link between the words of the action name and the words associated with that action is high.  Creating trainable, but semantically meaningless tokens reduces one's ability to perform zero-shot generalization and create new actions with meaningful names. Also, our focus was to provide a strong baseline that generalizes and not find the best model for the fixed action vocabulary task.
> - **BibTeX:**: Done
> - **Paper is good enough that a few rounds of iteration may solve these issues**: Thank you for the positive feedback. We have put in a lot of work over the past two weeks to address your excellent points, including: re-working the entire related work section and experimental results, and re-running experimentation to provide qualitative analysis. We also created a human performance evaluation as requested by other reviewers. We would be happy to engage in further rounds of iteration, but we do hope that we have managed to resolve most if not all of the key concerns through these changes.

---

> > ### Comment · Reviewer_ST5G · 2023-01-04
> > **Thank you for the many updates**
> >
> > I deeply appreciated the different changes made by the author. The paper is a lot better in this way. Congratulation. It looks good to me in this current stage.

---

### Decision · Action_Editors · 2023-01-15

**Recommendation:** Accept as is

**Comment:**

All the reviewers found the paper to be novel, interesting and well written and the empirical results to be quite convincing.
There were initial concerns around comparison to prior work and the paper framing but the authors have comprehensively revised the paper to address these issues and the current version of the paper is publication worthy.

Based on reviewer suggestions, I also recommend the Featured certification since this paper tackles the challenging task of dialog structure induction which has been a longstanding goal in dialog, plus has important applications in real-world settings. The paper is also well-written with comprehensive experiments and analyses, and is likely to substantially inform future work in this direction. As stated by one of the reviewers, "Workflow Discovery can be interesting to a broader audience since action-based dialogue is common in the real-world."

**Audience:**

This work introduces a new task and will be of substantial interest to the dialog community, with workflow discovery being an important yet challenging task with several practical implications as well.

**Claims And Evidence:**

This paper tackles the task of workflow discovery in dialogues where the goal is to predict dialog structure given dialog utterances. The authors propose a seq2seq approach and test it extensively on a recently released dataset that contains groundtruth workflows (ABCD). The results are quite promising and convincing, and the paper also compares the method on related previous tasks like action/dialog state tracking.

---

> ### Author Response · Authors · 2023-02-12
> **Thank you for the feedback**
>
> We want to thank all the reviewers for their time and comments. We believe that the feedback made our work stronger.